# Cellular determinants of parvovirus B19 susceptibility in the human placenta

Corinne Suter[1,2], Melanie Küffer[1], Jan Bieri[1], Amal Fahmi[3,4], David Baud[5,6], Marco P. Alves[3,4,7], Carlos Ros[1]*

1 Department of Chemistry, Biochemistry and Pharmaceutical Sciences, University of Bern, Bern, Switzerland, 2 Graduate School for Cellular and Biomedical Sciences, Bern, Switzerland, 3 Institute of Virology and Immunology, Bern, Switzerland, 4 Department of Infectious Diseases and Pathobiology, Vetsuisse Faculty, University of Bern, Bern, Switzerland, 5 Materno-Fetal and Obstetrics Research Unit, Mother-Woman-Child Department, University Hospital of Lausanne, Lausanne, Switzerland, 6 Faculty of Biology and Medicine, University of Lausanne, Lausanne, Switzerland, 7 Multidisciplinary Center for Infectious Diseases, University of Bern, Bern, Switzerland

* carlos.ros@unibe.ch

## Abstract

Parvovirus B19 (B19V) is a prevalent human pathogen that can cross the placenta by a mechanism that remains unknown, posing a risk of severe fetal complications, particularly during the first trimester of pregnancy. We investigated the expression of B19V-specific receptors in the three trophoblast cell types, cytotrophoblasts (CTBs), syncytiotrophoblasts (STBs), and extravillous trophoblasts (EVTs), and assessed their susceptibility to infection. VP1uR, the receptor that mediates viral uptake and infection in erythroid progenitor cells, is expressed in CTBs and STBs, but not in EVTs. Globoside, a glycosphingolipid that is essential for the escape of the virus from endosomes, is also expressed in these cells, except for choriocarcinoma-derived CTBs. In the latter, the absence of globoside can be overcome by promoting endosomal leakage with polyethyleneimine. While erythropoietin receptor (EpoR) signaling is associated with the strict erythroid tropism of B19V, it is not required for infection in trophoblasts. Transfection experiments revealed that highly proliferative first-trimester CTBs are more susceptible to B19V infection than the low-proliferative CTBs from term placenta. These findings demonstrate that B19V targets specific trophoblast cells, where viral entry and replication are collectively mediated by VP1uR, globoside, and high cellular proliferative activity, but are independent of EpoR signaling.

## Author summary

Parvovirus B19 is a common human pathogen that can cross the placenta and infect the fetus, though the mechanism remains unknown. Here, we demonstrate that specific trophoblast populations at the maternal-fetal interface create conditions that enable infection. Viral entry and replication depend on specific

Data availability statement: All data are in the manuscript and/or Supporting information files.

Funding: This study was supported by a grant from the Swiss National Science Foundation (grant 320030_207850 to C.S.) (https://portal.snf.ch/core/home). The funders had no role in study design, data collection and analysis, decision to publish, or preparation of the manuscript.

Competing interests: The authors have declared that no competing interests exist.

receptors expressed in the syncytiotrophoblasts and cytotrophoblasts in placental villi, as well as the high proliferative activity that is characteristic of the early stages of gestation. Unlike erythroid progenitor cells, trophoblast infection occurs independently of erythropoietin signaling. These findings broaden the known erythroid tropism of B19V to include trophoblast cells, providing a mechanistic explanation for the heightened vulnerability of the early placenta to B19V.

## Introduction

Parvovirus B19 (B19V) is a highly prevalent human pathogen responsible for erythema infectiosum, a pediatric disease marked by a characteristic erythematous rash and mild systemic symptom. The virus exhibits a pronounced tropism for erythroid progenitor cells (EPCs) within the bone marrow, where it establishes a lytic infection that disrupts normal erythropoiesis [1]. The lytic infection in EPCs leads to a transient interruption of erythropoiesis, which leads to the hematological disorders associated with the infection, particularly in individuals with chronic hemolytic anemia or immunosuppression. Infection during pregnancy is frequently associated with perinatal complications [2]. Despite its high prevalence and impact on human health, B19V remains a neglected viral infection, with no vaccines or antiviral treatments currently available.

The 25-nm capsid of B19V is assembled from only two proteins, VP1 and VP2 [3]. The VP1 protein contains an N-terminal extension, termed the VP1 unique region (VP1u), which includes the receptor-binding domain (RBD) [4,5]. The molecular entity or complex targeted by the VP1u RBD that mediates viral binding and uptake, referred to as VP1uR, is a highly restricted receptor that has so far been exclusively found in the target EPCs in the bone marrow, consistent with the marked erythroid cell tropism of the virus [6,7]. Transmission through the respiratory route is not mediated by VP1uR. Instead, the virus engages globoside, a glycosphingolipid expressed on the surface of the ciliated epithelial cells. The interaction with globoside is tightly regulated by pH [8]. Within the acidic environment of the nasal mucosa, the virus binds to globoside on the apical surface of epithelial cells. Subsequently, the virus is transported across the cell via transcytosis and released at the basolateral side, where the neutral pH facilitates its dissociation from globoside [9]. In addition to mediating viral translocation across the respiratory epithelium, globoside has an additional critical role. Following VP1uR-mediated endocytosis in EPCs, viral particles accumulate in endosomes, where low pH induces dissociation from VP1uR and binding to globoside, a critical interaction that facilitates escape from the endosomal compartment [8,10].

The acute phase of infection is typically associated with exceptionally high-titer viremia, which is unparalleled by any other viral infection. During this highly viremic phase, the virus can invade the placenta and reach the fetus [11,12]. Transplacental transmission can result in severe complications, including fetal anemia, spontaneous miscarriage, hydrops fetalis, and intrauterine demise [13,14]. Early- to mid-pregnancy

maternal infection markedly increases the risk of adverse effects on the fetus [15–17]. The increased risk in early gestation is attributed to the high expression of globoside in trophoblasts during this period [18]. Between 30–50% of women that become pregnant are nonimmune and susceptible to B19V infection [19–21]. In North America, between 1% and 2% of pregnant individuals become infected, with rates surging to around 10% during epidemic periods [21]. Among those with acute infection, transplacental transmission to the fetus has been documented in 17–51% of cases [22–24]. It has been estimated that approximately 3% of first-trimester spontaneous abortions may be due to B19V infection [25]. Pregnant women infected with B19V require careful monitoring, including serial ultrasounds to detect fetal anemia and hydrops fetalis. Early detection allows for timely interventions, such as intrauterine transfusions, to manage fetal anemia [21].

Several studies have examined the interaction between B19V and the human placenta. Placental tissues from B19V infected women showed an increase in CD3-positive T cells and interleukin-2 production, suggesting that the infection triggers an immune response in the placenta [26]. The virus induces apoptosis in trophoblasts, which may contribute to placental dysfunction and adverse pregnancy outcomes [27]. Additionally, B19V has been linked to the deregulation of apoptotic pathways in placental tissues [28]. Empty capsids of B19V have been shown to bind to villous trophoblast cells via the globoside receptor, suggesting a role for globoside in facilitating viral transmission across the maternal-fetal interface [29]. A study reported a potential association between maternal B19V infection and placental abruption, suggesting that virus-induced apoptosis in trophoblasts may contribute to placental damage [30]. While these studies reveal the association between the virus and adverse pregnancy outcomes, none of them provide conclusive evidence of parvovirus B19 infection within the placenta, nor do they clarify the specific receptors involved or the cellular tropism of the virus among the different trophoblast populations.

Limited knowledge of B19V-host interactions at the maternal-fetal interface, combined with the lack of vaccines or antiviral therapies, places infected pregnant individuals at significant risk. Identifying the cellular determinants that regulate trophoblast permissiveness to B19V is crucial for advancing our understanding of the mechanisms driving vertical transmission. To address this, we employed trophoblast models representing the specialized trophoblast populations of the human placenta. Our findings reveal that VP1uR, the highly restricted B19V receptor known to mediate viral uptake and infection in EPCs, is also expressed in trophoblasts, identifying a previously unrecognized susceptibility factor at the maternal-fetal interface. In addition to VP1uR, globoside expression and increased cellular proliferation were revealed as major factors governing B19V infection in trophoblasts.

## Results

### Placental trophoblasts support VP1u binding and uptake

To assess the presence of the cellular determinant mediating VP1u binding and uptake, referred to here for simplicity as VP1uR, we employed C-terminally and N-terminally truncated VP1u constructs. The C-terminally truncated construct (ΔC128) retains the functional receptor-binding domain (RBD; aa 5–80), while the N-terminally truncated construct (ΔN29) carries a disrupted, non-functional RBD and served as a negative control (Fig 1A). These FLAG-tagged recombinant VP1u proteins were incubated with UT7/Epo, a human megakaryoblastoid cell line that expresses VP1uR [5]; BeWo and JEG-3 cells, first-trimester choriocarcinoma-derived cytotrophoblasts (CTBs); hPTC^CTB, CTBs derived from term placenta [31]; and HTR-8/Svneo, a model of extravillous trophoblasts (EVTs) [32]. REH cells, a B-cell precursor leukemia line lacking VP1uR expression [6], served as a negative control. After 30 minutes of incubation at 37°C, cells were fixed and stained with an anti-FLAG antibody to detect internalization. The VP1u construct containing the functional RBD (ΔC128) was detected in all trophoblast cell types except HTR-8/SVneo. The VP1u construct with the non-functional RBD (ΔN29) showed no binding (Fig 1B). These findings reveal, for the first time, the expression of a VP1u-binding receptor beyond the erythroid lineage.

Next, a virus-like particle (VLP)-based binding assay was used. The full-length B19V VP1u region was chemically conjugated to purified MS2 bacteriophage VLPs via click chemistry and labeled with Atto 488 for fluorescence detection.

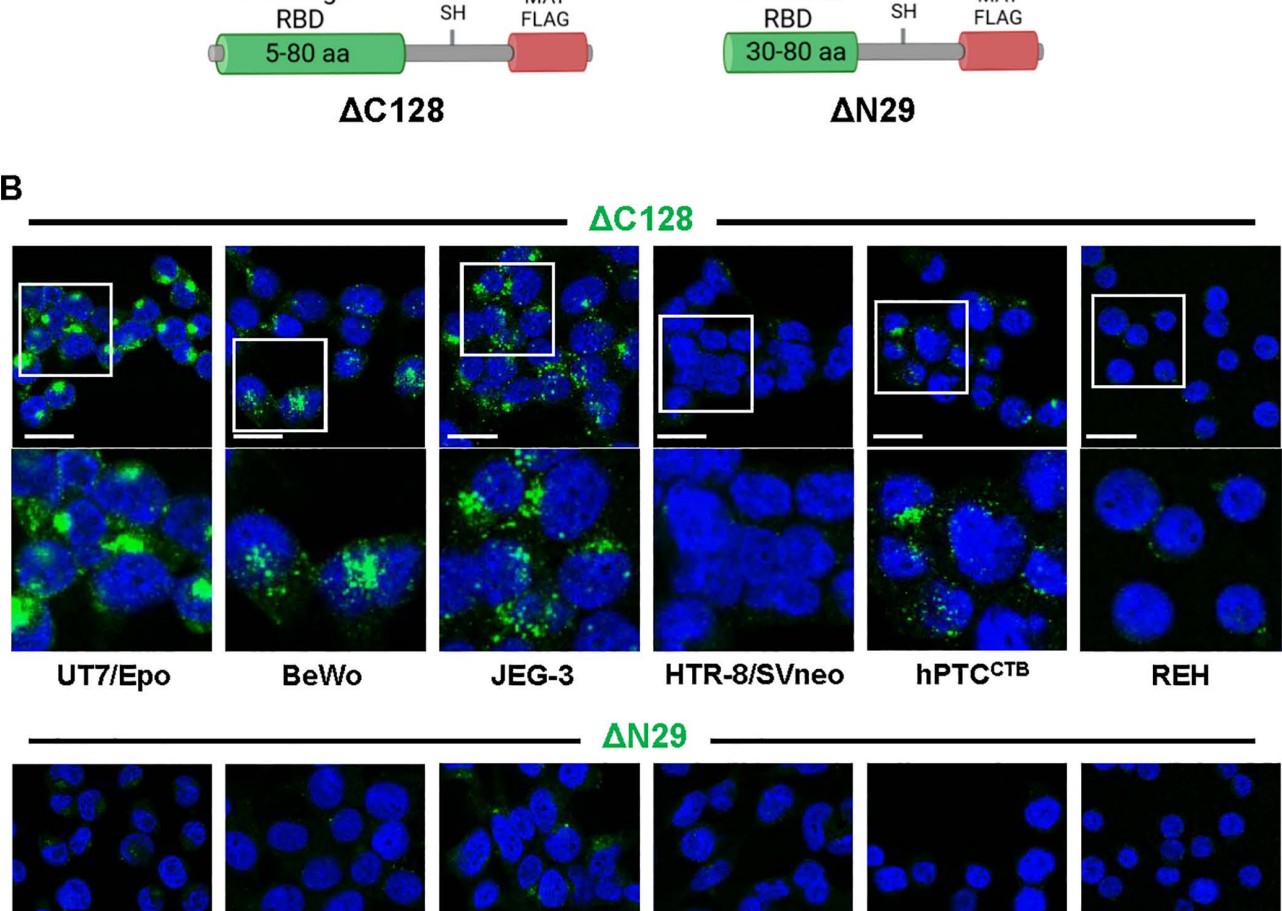

**Fig 1. Detection of VP1uR expression in human placental trophoblasts. (A)** Schematic representation of recombinant VP1u constructs. ΔC128 harbors a complete (5-80 aa) receptor-binding domain (RBD). ΔN29 harbors a truncated and non-functional RBD. The VP1u constructs include a cysteine-derived thiol group (-SH) for disulfide bond-mediated dimerization, an N-terminal MAT tag for efficient expression and solubility, and a C-terminal FLAG tag for detection with anti-FLAG antibodies. Created in https://BioRender.com. **(B)** Trophoblast cells were incubated at 37°C for 30 min with recombinant VP1u constructs and visualized by confocal microscopy with an anti-FLAG antibody (green). UT7/Epo cells served as positive and negative control, respectively. DAPI (blue). Scale bar, 20 μm. Created in BioRender. Ros, **C.** (2026) https://BioRender.com/b5rm6vq.

The use of VP1u-decorated MS2 VLPs allowed assessment of whether VP1u alone is sufficient to mediate the uptake of a virus-sized particle, independent of other B19V capsid domains or additional cellular factors. Since unmodified MS2 VLPs lack intrinsic affinity for trophoblasts, Atto 488-labeled MS2 VLPs without VP1u served as a negative control (Fig 2A). MS2 VLPs constructs were incubated with the various trophoblast subtypes at 37°C for 30 minutes to allow receptor-mediated binding and internalization. UT7/Epo cells served as a positive control. Consistent with the recombinant VP1u binding assay, VP1u-coated MS2 VLPs bound to and were internalized by all trophoblast cell types tested, with the exception of HTR-8/SVneo. No binding was detected with Atto 488-labeled MS2 VLPs lacking VP1u (Fig 2B). These results confirm the presence of functional VP1uR in CTBs from various placental origins and its absence in EVTs.

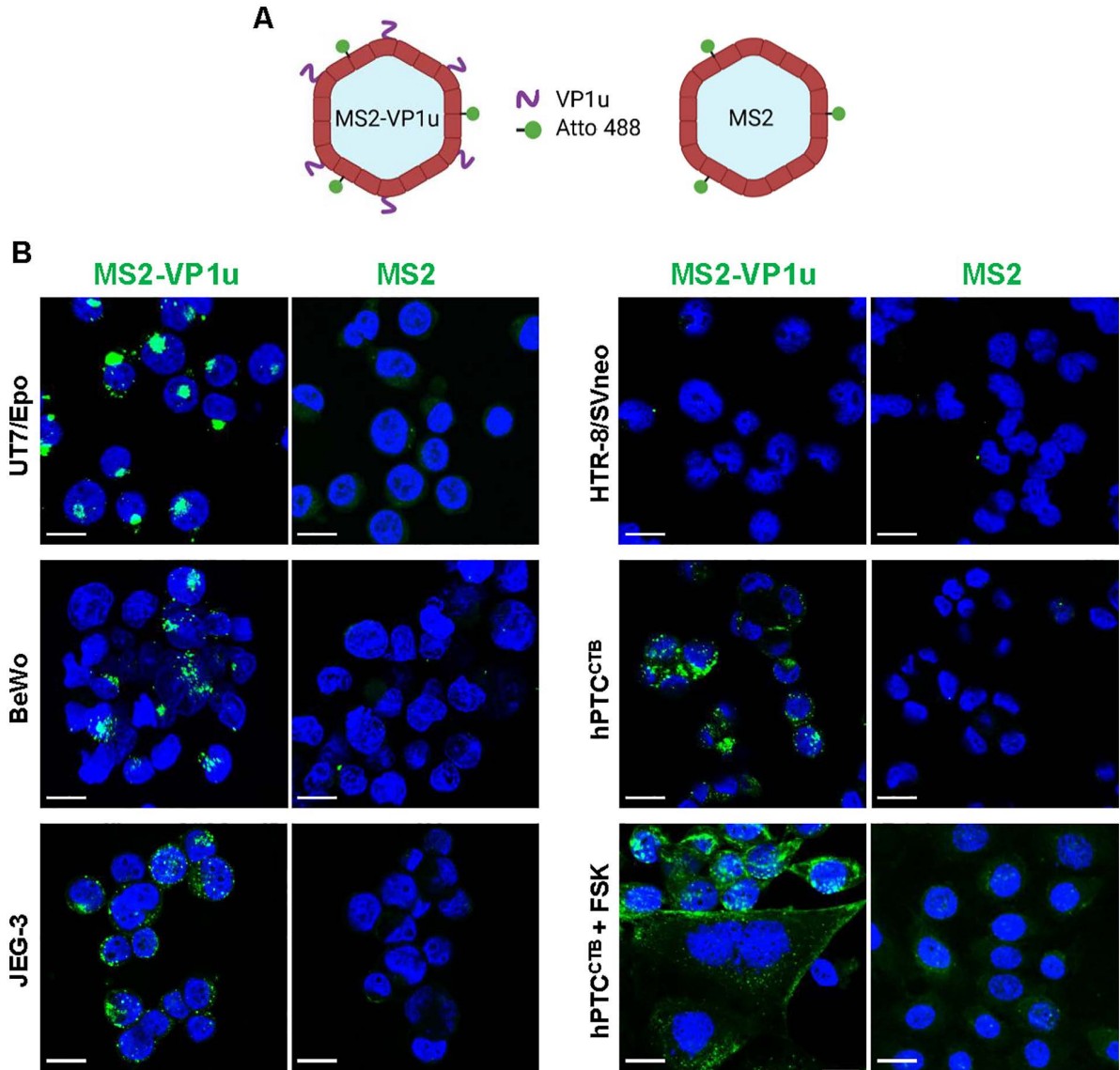

**Fig 2. VP1uR is functional and mediates viral particle internalization in trophoblasts. (A)** Schematic representation of MS2 VLPs conjugated with VP1u and/or Atto 488 by click chemistry. Created in https://BioRender.com. **(B)** UT7/Epo and different trophoblast subtypes were incubated at 37°C for 30 min with MS2 constructs and visualized by confocal microscopy. hPTCCTB cells were treated with forskolin (hPTCCTB + FSK) to induce differentiation into STBs. Following differentiation, cells were incubated at 37°C for 30 min with MS2 constructs and visualized by confocal microscopy. MS2-VLPs (green). DAPI (blue). Scale bar, 20 μm. Created in BioRender. Ros, **C.** (2026) https://BioRender.com/q40s7au.

Forskolin induces the differentiation of CTBs into syncytiotrophoblasts (STBs) by activating adenylyl cyclase, which promotes cell fusion [33]. hPTCCTB cells were differentiated into STBs using forskolin. STB differentiation was confirmed by the formation of large multinucleated syncytia. Fluorescence microscopy using MS2-VP1u VLPs showed that VP1uR expression was maintained following the differentiation of CTBs into STBs (Fig 2B).

Given that the AXL receptor tyrosine kinase (AXL) has been proposed as a potential receptor mediating VP1u binding in erythroid cells [34], we investigated whether its expression overlaps with VP1uR in placental trophoblasts. AXL mRNA expression was analyzed across trophoblast cell lines (S1A Fig), and the specificity of the RT-PCR products was

confirmed by agarose gel electrophoresis (S1B Fig). The expression of AXL was also analysed by immunofluorescence microscopy with a specific antibody (S1C Fig). Our analysis revealed no consistent co-expression across trophoblast subtypes: BeWo and JEG-3 cells express VP1uR but not AXL; HTR-8/SVneo cells express AXL but lack VP1uR; and hPTC[CTB] cells express both receptors.

## Globoside expression in trophoblast cell lines

Globoside is essential for B19V infection, enabling transcytosis through the respiratory epithelium [9], and facilitating endosomal escape of incoming virions in EPCs [10]. Given its essential role, we examined globoside expression across the different trophoblast subtypes to assess their potential susceptibility to B19V infection. The mRNA levels of globoside synthase (Gb4) and globotriaosylceramide synthase (Gb3), were determined by quantitative RT-PCR (RT-qPCR) using specific primers for β3GalNT1 (Gb4) and A4GalT (Gb3) mRNAs and normalized to GAPDH. UT7/Epo cells, which express globoside and are commonly used to study B19V infection, served as a positive control. Globoside knockout UT7/Epo cells, generated by CRISPR/Cas9-mediated knockout in a previous study [35], and REH cells, which naturally lack globoside expression, were used as negative controls. The results showed that BeWo and JEG-3 cells expressed Gb3 synthase but lacked Gb4 synthase. In contrast, HTR-8/SVneo and hPTC[CTB] cells expressed both Gb3 and Gb4 synthases (Fig 3A). Immunofluorescence staining with a globoside-specific antibody confirmed the RT-qPCR results, showing that BeWo and JEG-3 cells lacked detectable globoside expression, whereas it was present in HTR-8/SVneo and hPTC[CTB] cells (Fig 3B).

To examine whether globoside expression is maintained during the differentiation of CTBs into STBs, we induced syncytialization in hPTC[CTB] cells using forskolin. Syncytialization induced with forskolin has been reported to alter the composition and abundance of the placental glycocalyx, potentially impacting glycosphingolipid expression [36]. Following STB differentiation, Gb4 expression remained detectable, indicating that globoside expression is maintained in STBs (Fig 3B). Consistently, the mRNA levels of Gb3 and Gb4 synthases remained unchanged before and after differentiation, indicating that syncytialization does not affect globoside expression (Fig 3C). Furthermore, forskolin-induced syncytialization of BeWo cells did not restore globoside expression in these cells (S2A Fig).

Considering that oxygen tension is low during early pregnancy and critically regulates gene expression [37], and that Epo (erythropoietin) signaling through the Epo receptor (EpoR) in trophoblasts can influence differentiation and survival [38], we investigated whether globoside expression is modulated by oxygen levels and/or Epo stimulation. BeWo cells were cultured under normoxic or hypoxic conditions, with or without Epo. While Gb3 expression remained consistent across all conditions, globoside expression was consistently undetectable, irrespective of oxygen tension or Epo stimulation (S2B and S2C Fig).

The expression of globoside was evaluated in cryosections of term placental tissue. To verify the presence of trophoblast subpopulations, sections were stained with BCL-2, Trop-2, and CD138. BCL-2 is used to label syncytiotrophoblast, Trop-2 identifies both cytotrophoblasts and syncytiotrophoblasts, and syndecan-1 marks the syncytiotrophoblast apical surface [39,40]. These markers together allow reliable delineation of trophoblast compartments in placental tissue sections. Staining with an antibody against globoside revealed a scattered signal distributed along the periphery of the villi and, more prominently, within the villous core. This pattern suggests that globoside expression is not restricted to the trophoblast layers but is more abundant in non-trophoblast cell populations residing within the villous core (S3 Fig). These findings corroborate previous studies that have reported widespread expression of globoside in placental trophoblasts [18,41,42]. The absence of globoside expression in the choriocarcinoma cell lines BeWo and JEG-3 is likely attributable to their malignant nature, as malignant transformation is known to alter glycosyltransferase expression [43,44].

## B19V uptake by placental trophoblasts correlates with VP1uR expression

To assess the functional relevance of VP1uR expression in trophoblast cells, a viral internalization assay was performed. UT7/Epo cells were included as a positive control, and REH cells as a negative control to establish background signal.

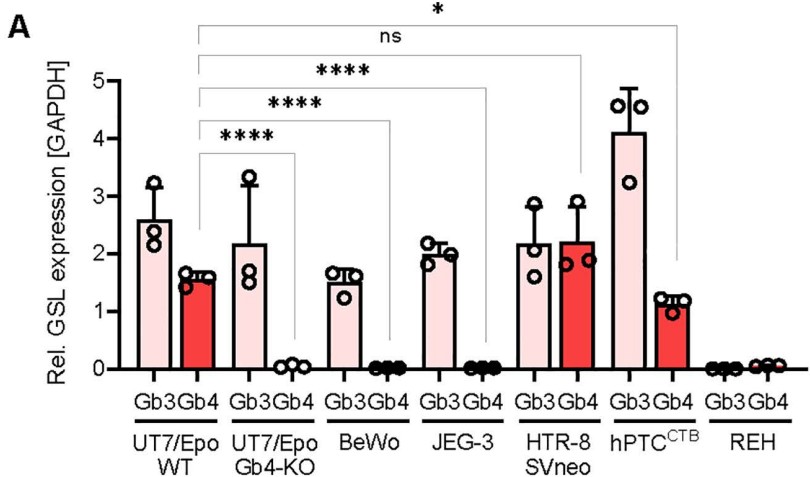

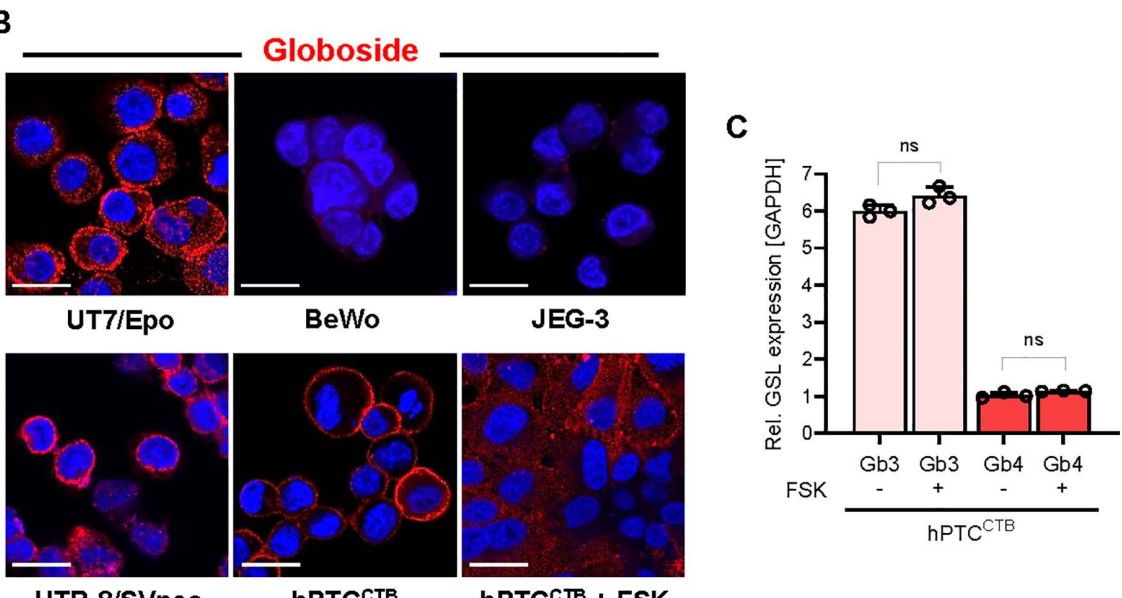

**Fig 3. Globoside expression in different trophoblast subtypes. (A)** Relative expression of Gb3 (Gb3 synthase) and Gb4 (Gb4 synthase) mRNA in different trophoblasts. mRNA levels were measured by RT-qPCR and normalized to GAPDH mRNA. GSL, glycosphingolipid. **(B)** Immunostaining of UT7/Epo and trophoblasts with an anti-globoside antibody (red). DAPI (Blue). Scale bar, 20 μm. **(C)** Relative expression of Gb3 and Gb4 mRNA in hPTC$^{CTB}$ cells following differentiation into STBs using forskolin (FSK). mRNA levels were measured by RT-qPCR and normalized to GAPDH mRNA. All results are presented as the mean ± SD of three independent experiments. Statistical significance was calculated using two-sided Student's t-test. *$p < 0.05$; ****$p < 0.0001$; ns, non-significant.

After 1 h incubation with B19V at 37°C, residual surface-bound virus was removed by trypsinization, and intracellular viral genomes were quantified by qPCR. B19V DNA was readily detected in BeWo, JEG-3, and hPTC$^{CTB}$ cells, all of which express VP1uR, whereas no internalization above background was observed in VP1uR-negative HTR-8/SVneo and REH cells (Fig 4A).

To corroborate these findings, immunofluorescence staining for B19V capsid proteins was performed. A strong viral signal was observed in BeWo cells, weaker staining in JEG-3, and no detectable signal in HTR-8/SVneo and hPTC$^{CTB}$ (Fig 4B). The relative signal intensity mirrored the qPCR data, and notably, these differences in viral uptake were consistent

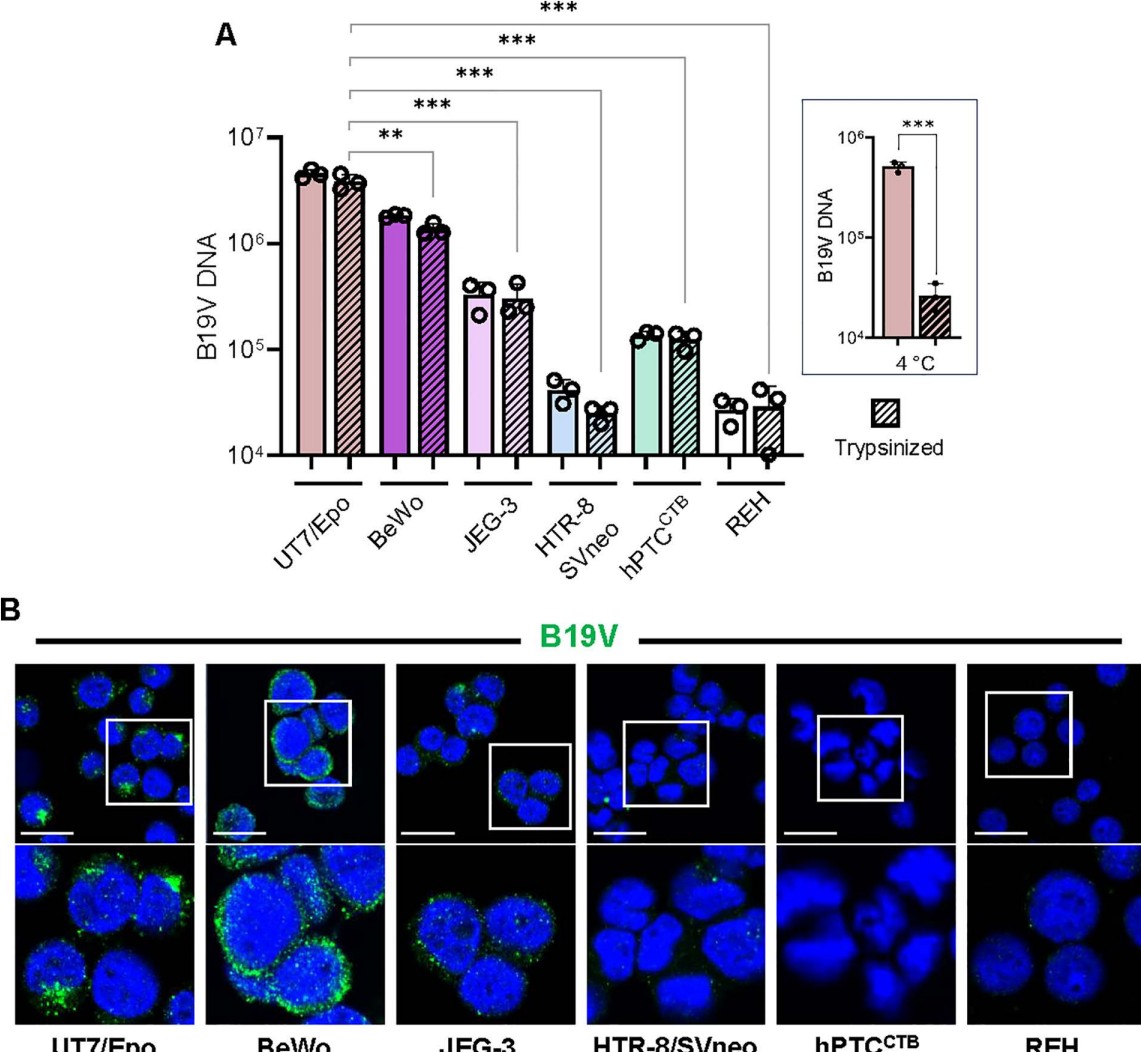

**Fig 4. B19V uptake in trophoblasts correlates with VP1uR expression. (A)** Quantification of cell-associated B19V genomes after incubation, with or without trypsin treatment to distinguish internalized from surface-bound virus. UT7/Epo cells served as a positive control, and REH cells as a negative control to define background signal. The graph shown on the right depicts the trypsin control, where UT7/Epo cells were incubated with B19V at 4 °C to prevent viral uptake prior to trypsin treatment. Two-sided Student's t-test. The results are presented as the mean ± SD of three independent experiments. **$p < 0.01$; ***$p < 0.001$. **(B)** Immunofluorescence images showing B19V uptake in UT7/Epo and trophoblast cells. Cells were incubated with B19V ($10^5$ genome equivalents per cell) at 37°C for 30 min. After incubation, cells were washed, fixed and stained with a monoclonal antibody against capsids (mAb 521, green). DAPI (Blue). Scale bar, 20 µm.

with the uptake patterns observed for recombinant VP1u and VP1u-coated MS2 in these cells (Figs 1 and 2). Importantly, B19V uptake was detected exclusively in cells expressing VP1uR, highlighting its essential role in mediating viral entry in trophoblasts.

## B19V infection across trophoblast types and gestational stages

Following B19V internalization in trophoblast cells, the expression of the early viral transcript NS1 was analyzed 24 h and 48 h post-infection. In BeWo and JEG-3, NS1 mRNA levels were in the order of three logs lower than in UT7/Epo cells,

consistent with the lack of Gb4 expression in these cell lines, which is essential for endosomal escape [10]. Similarly, HTR-8/SVneo cells, which do not express VP1uR, showed minimal NS1 expression. The low viral mRNA levels detected in BeWo, JEG-3, and HTR-8 likely reflect inefficient viral entry via alternative mechanisms, resulting in transcription near background levels. In contrast, hPTC$^{CTB}$ cells, which express VP1uR and Gb4, exhibited significantly higher NS1 expression, though still lower than levels observed in UT7/Epo cells (Fig 5A).

Progression of infection beyond the early phase was evaluated by quantifying mRNA levels of the structural viral proteins VP1 and VP2 using RT-qPCR. Consistent with low or absent expression of NS1, VP1/2 transcripts were undetectable in BeWo, JEG-3, and HTR-8/SVneo cells, with values comparable to the input control. In contrast, VP1/2 mRNA was detected in hPTC$^{CTB}$ cells, indicating that infection had progressed beyond the early transcriptional stage (Fig 5B).

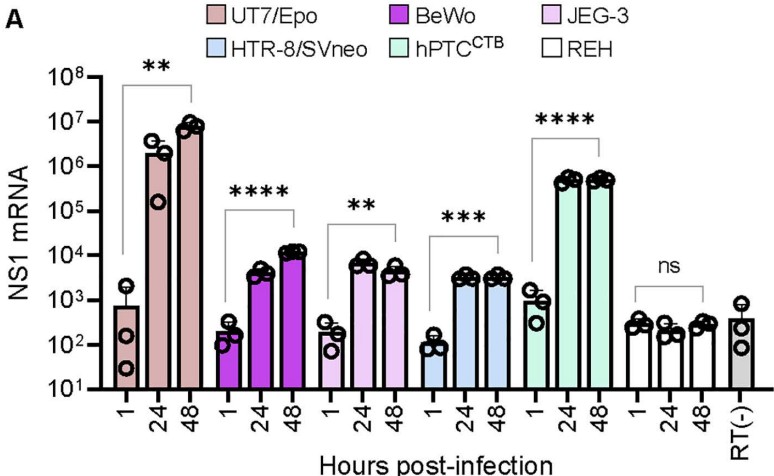

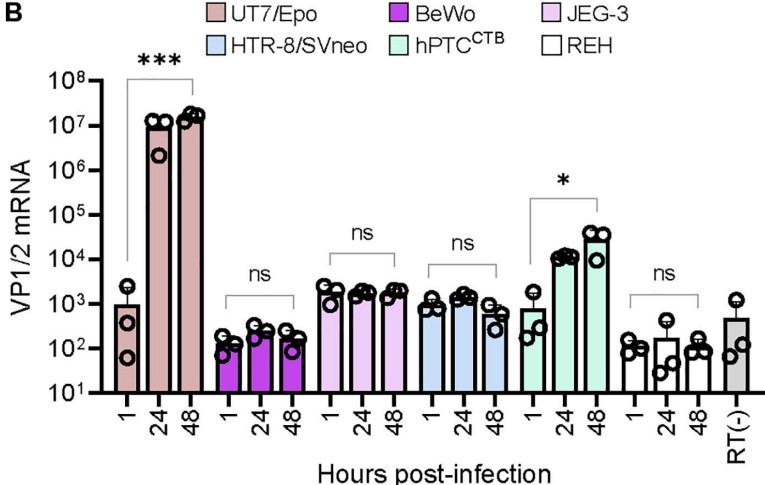

**Fig 5. Expression of B19V non-structural and structural mRNAs in trophoblast cells. (A)** Quantification of B19V infection in different trophoblast subtypes. UT7/Epo and REH cells were used as positive and negative controls, respectively. Cells were infected with B19V ($10^5$ genome equivalents per cell) at 37°C for 30 minutes. NS1 mRNA levels were quantified by RT-PCR at 1 h (input), 24 h and 48 h post-infection. **(B)** VP1/VP2 mRNA levels were quantified in the same experimental setup to assess structural viral gene expression. All results are presented as the mean±SD of three independent experiments. Statistical significance was calculated using two-sided Student's t-test. *$p<0.05$; **$p<0.01$; ***$p<0.001$; ****$p<0.0001$; ns, non-significant.

However, VP1/2 transcript levels in hPTC$^{CTB}$ remained substantially lower than those observed in UT7/Epo cells, consistent with inefficient viral DNA replication and lack of productive infection.

B19V infection requires both VP1uR and Gb4 for cellular entry and depends on active cell proliferation for successful replication. Despite a robust NS1 expression, the limited expression of VP1/VP2 mRNAs observed in hPTC$^{CTB}$ may be attributed, at least in part, to the reduced proliferative capacity and increased cellular senescence of CTBs during late gestation [45]. To investigate this possibility, we assessed the proliferation rates of different trophoblast subtypes by quantifying β-actin gene expression over a four-day culture period. As expected, the results confirmed that first-trimester trophoblasts exhibit high proliferative activity, whereas hPTC$^{CTB}$ derived from term placenta show the lowest proliferation rate (S4 Fig).

Together, these results demonstrate that none of the trophoblast cell types analyzed are permissive to productive B19V infection, as each lack at least one essential determinant, i.e., VP1uR, globoside, or sufficient proliferative activity.

### Transcriptional activity of B19V genomes following transfection in trophoblast subtypes

To assess the activity of the B19V P6 promoter in different trophoblast subtypes independently of viral entry, transfection assays were conducted using purified single-stranded viral DNA genomes. NS1 and VP1/2 mRNA levels were quantified by RT-qPCR two days post-transfection. Each cell type was assessed for transfection efficiency by introducing a GFP-expressing plasmid and quantifying the proportion of GFP-positive cells via fluorescence microscope. UT7/Epo cells were used as reference control. Transfection efficiency varied among the trophoblast cell lines: HTR-8/SVneo exhibited the highest efficiency (4.5-fold higher than UT7/Epo), followed by BeWo (3.5-fold). In contrast, JEG-3 and hPTC$^{CTB}$ showed transfection rates comparable to those of UT7/Epo cells (Fig 6A). Subsequent viral mRNA measurements reflect population-level transcription and were not normalized to transfection efficiency. Remarkably, NS1 mRNA quantification revealed that first-trimester CTBs, specifically JEG-3 and BeWo cells, exhibited significantly higher NS1 expression levels, showing 42-fold and 12-fold increases, respectively, compared to UT7/Epo cells (Fig 6B), consistent with a strong P6 promoter activity in these cells.

Accurate measurement of viral DNA replication is not possible after transfection due to the presence of input viral genomes. Given that robust DNA replication is required for the expression of the structural genes VP1 and VP2, their detection was therefore used as a replication-dependent readout. Consistent with the NS1 mRNA data, quantification of VP1/VP2 mRNA also showed a similar trend in first-trimester CTBs (Fig 6C). Notably, the substantial increases in NS1 and VP1/VP2 mRNA expression observed in BeWo and JEG-3 cells far exceeded the differences in transfection efficiency. In contrast, although HTR-8/SVneo cells exhibited the highest transfection efficiency, they did not show a corresponding increase in viral gene expression.

To determine whether the elevated NS1 mRNA levels resulted in NS1 protein accumulation, NS1 expression was examined by immunofluorescence using a specific anti-NS1 antibody. Consistent with the transcriptional data, strong NS1 protein signals were detected in BeWo and JEG-3 cells two days post-transfection (Fig 6D). These data demonstrate that high P6-driven transcription in first-trimester CTB-derived cell lines is accompanied by robust NS1 protein accumulation.

Overall, the findings indicate that the first-trimester CTBs support higher levels of P6 promoter-driven viral transcription than that of EVTs or CTBs derived from term placentas. As previously suggested, the reduced proliferative capacity of term placental CTBs likely contributes to their lower levels of viral gene expression, highlighting the critical role of cellular proliferation in determining B19V infection. Although the first-trimester CTB-derived cell lines, BeWo and JEG-3 exhibited high P6-driven transcriptional activity, the absence of globoside expression blocks viral entry, rendering these cells non-permissive for productive B19V infection.

### PEI rescues B19V infection in trophoblasts lacking globoside

Globoside plays a pivotal role in the endosomal escape of incoming B19V particles, facilitating their release from endosomes into the cytoplasm, which is a crucial step for viral replication [10]. Several studies have shown that primary

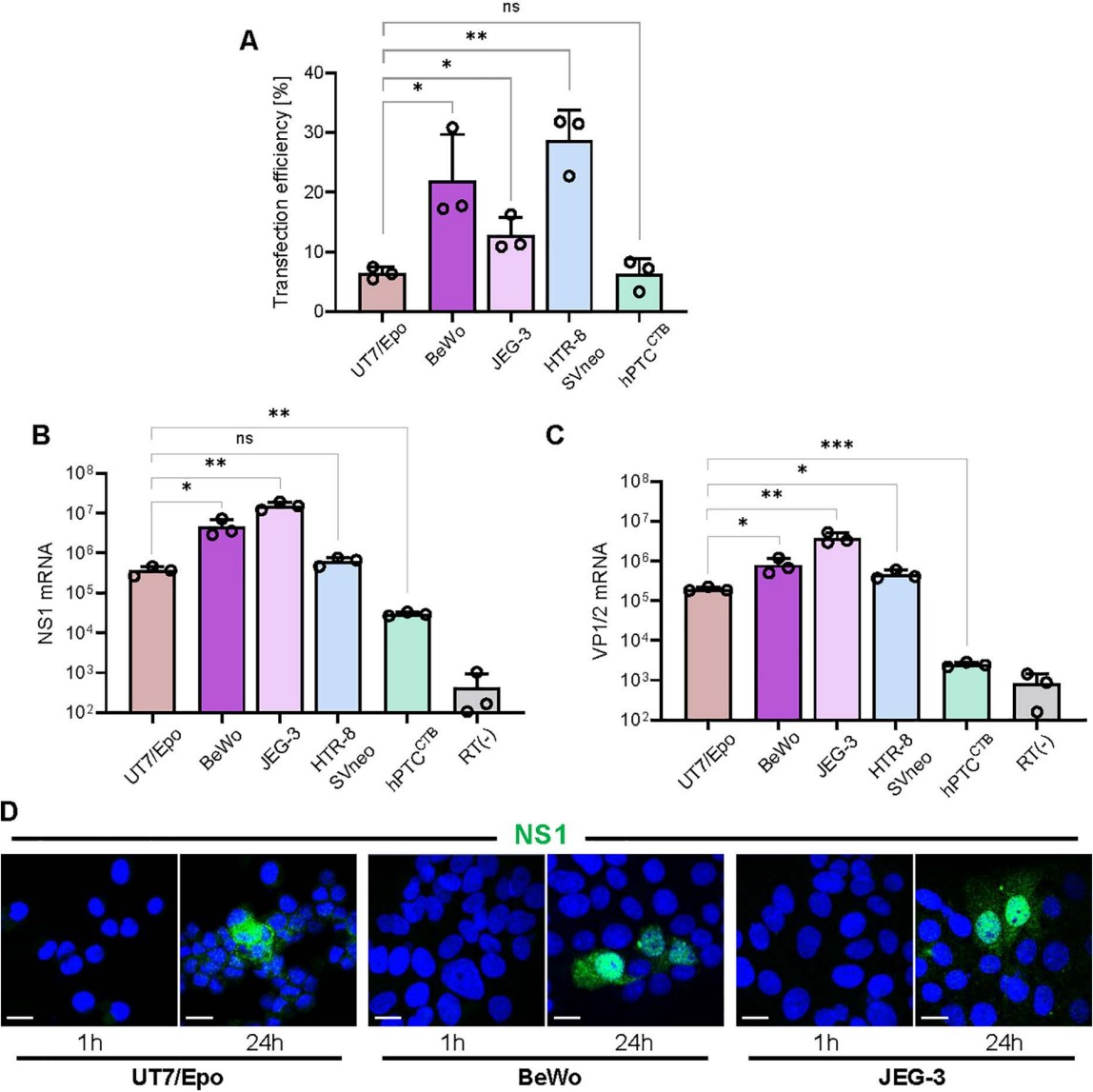

**Fig 6. Transcriptional activity of B19V genomes following transfection in trophoblast subtypes. (A)** Transfection efficiency was evaluated with a plasmid encoding green fluorescent protein (GFP). Cells were visualized by fluorescence microscopy, and GFP-positive cells were quantified as a percentage of total cells. **(B)** Purified single-stranded B19V genomes were transfected into the indicated cell types. UT7/Epo cells were included as a permissive reference. At 24 h post-transfection, NS1 mRNA expression was quantified by RT-PCR to assess early viral gene expression. **(C)** Under the same experimental conditions as in **(B)**, VP1/VP2 mRNA levels were quantified by RT-PCR to evaluate structural gene expression following transfection. **(D)** Detection of NS1 protein by immunofluorescence using an NS1-specific antibody (mAb 1424; green) in transfected BeWo and JEG-3 cells at 1 h (input) and 24 h post-transfection. UT7/Epo cells were included as a positive control. DAPI (Blue). Scale bar, 20 μm. All results are presented as the mean ± SD of three independent experiments. Statistical significance was calculated using two-sided Student's t-test. *$p < 0.05$; **$p < 0.01$; ***$p < 0.001$; ns, non-significant.

trophoblasts, whether isolated from placentas or analyzed in tissue sections, consistently express globoside [18,41,42]. Our findings align with this observation, showing that all trophoblast subtypes tested expressed globoside, with the exception of the two choriocarcinoma-derived BeWo and JEG-3 cell lines, which likely lack it due to tumor-associated changes in glycosphingolipid synthesis [43,44].

To circumvent the absence of globoside and promote endosomal escape, we employed polyethyleneimine (PEI), a cationic polymer known to enhance endosomal rupture by disrupting endosomal membranes and facilitating the release of internalized particles into the cytoplasm [46]. To evaluate the effect of PEI on endosomal integrity, BeWo and JEG-3 cells were incubated with tetramethylrhodamine-labeled dextrans (3 kDa), which serve as markers of endosomal compartmentalization. In untreated cells with intact endosomes, the rhodamine-labeled dextrans remained confined within discrete vesicles. In contrast, PEI-treated cells (≥1 µM) exhibited diffuse cytoplasmic fluorescence, indicating loss of endosomal integrity (Fig 7A and 7B).

Cells were infected with B19V in the presence of PEI (1 µM or 5 µM) at 37°C for 30 min. After removing unbound virus and PEI by washing, the infection was allowed to proceed for 24 h, and NS1 mRNA levels were quantified by RT-qPCR. The results revealed a dose-dependent increase in NS1 mRNA levels in PEI-treated cells, indicating enhanced viral replication (Fig 7C).

To confirm that the transcribed NS1 mRNA was effectively translated into protein, the presence of NS1 protein was examined by immunofluorescence microscopy with an antibody against B19V NS1 (mAb 1424) [47]. NS1 protein accumulation was detected in BeWo cells after 24 h and 48 h post-infection, confirming translation of the NS1 mRNA (Fig 7D). Given that NS1 is essential for viral DNA replication and the subsequent expression of structural proteins, the expression of VP1/2 mRNA was tested. RT-PCR confirmed the presence of VP1/VP2 mRNAs in PEI-treated BeWo cells, which correlated with the observed NS1 protein expression (Fig 7E). These results provide compelling evidence that PEI facilitates endosomal escape of incoming viruses, thereby circumventing the lack of globoside in BeWo cells.

## EpoR signaling is not required for B19V infection of trophoblasts

EpoR is a cell surface receptor that binds Epo, a hormone known for its role in erythropoiesis. Although EpoR signaling is not required for virus entry, it has been shown to be essential for B19V replication in EPCs and is thought to be a key factor underlying the pronounced tropism of B19V for human erythroid progenitors [48,49].

Both Epo and EpoR are expressed by trophoblasts in the human placenta, where EpoR signaling contributes to critical processes such as cell survival, angiogenesis, placental development, and adaptation to hypoxic conditions [38,50]. Although Epo expression has been observed in BeWo cells [51], quantitative data is not available, and expression in other trophoblast-derived cell lines, such as JEG-3, has not been assessed. To address this gap, we quantified Epo mRNA levels in BeWo and JEG-3 cells under normoxic and hypoxic conditions, using HepG2 and UT7/Epo cells as positive and negative controls, respectively. Epo mRNA was detectable in both trophoblast lines, albeit at substantially lower levels than in HepG2 cells, indicating low, nearly undetectable endogenous Epo expression (Fig 8A and 8B). Given the low baseline expression, we examined whether exogenous Epo supplementation could modulate BeWo cell proliferation or survival. However, supplementation with exogenous Epo had no significant impact on BeWo cells, in contrast to UT7/Epo cells, where exogenous Epo is essential for both cell proliferation and viability (Fig 8C). Transfection of BeWo cells with B19V DNA genomes, either in the presence or absence of exogenous Epo, revealed no substantial differences in NS1 and VP expression, suggesting that EpoR signaling is not essential for B19V infection in trophoblasts (Fig 8D).

Both BeWo and JEG-3 cells exhibited low but detectable levels of Epo expression, raising the possibility that even minimal EpoR signaling might be sufficient to support B19V infection. JAK2 is a central mediator of EpoR signaling, activating downstream STAT5 and MEK/ERK pathways, and its inhibition with AG490 has previously been shown to disturb B19V infection in erythroid progenitor cells [48]. AG490 is a tyrphostin compound that blocks JAK2 phosphorylation by competing with ATP [52]. At concentrations below 20 µM, AG490 had no significant effect on the proliferation or viability of BeWo cells (Fig 8E). To assess the impact of JAK2 inhibition on B19V gene expression, BeWo cells were treated with 5 or 20 µM AG490 prior to transfection with full-length B19V genomes. Subsequent quantification revealed that NS1 and structural VP

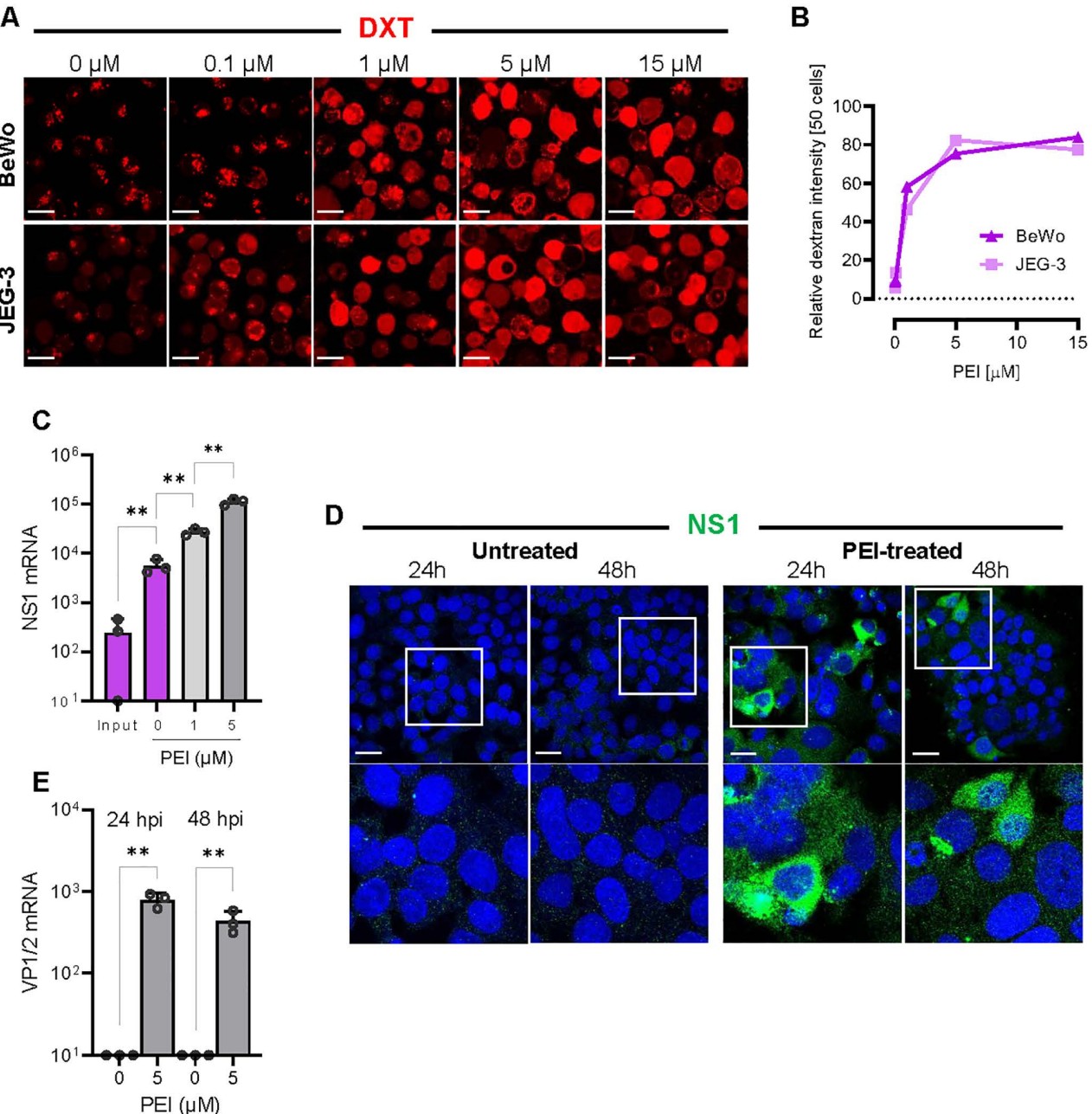

**Fig 7. PEI-mediated endosomal escape rescues B19V infection in globoside-deficient trophoblasts. (A)** Immunofluorescence images showing intracellular distribution of fluorescent dextrans (DXT: red) in BeWo and JEG-3 cells following treatment with increasing concentrations of PEI. Diffuse cytoplasmic localization of dextrans indicates release from endosomes and disruption of endosomal membranes. Scale bar, 20 μm. **(B)** Quantification of cells with cytosolic distribution of dextrans in BeWo and JEG-3 cells in response to PEI treatment. Fluorescence intensity was measured in 50 cells per condition using ImageJ. **(C)** Quantification of NS1 mRNA levels at 24 h post-infection in BeWo cells treated or untreated with PEI. **(D)** Detection of NS1 protein by immunofluorescence using an NS1-specific antibody (mAb 1424; green) in infected BeWo cells at 24 h and 48 h post-infection, with or without 5 μM PEI treatment. Scale bar, 30 μm. **(E)** Quantification of structural VP1/VP2 mRNA levels at 24 h and 48 h post-infection in BeWo cells treated or untreated with 5 μM PEI. All results are presented as the mean ± SD of three independent experiments. Statistical significance was calculated using two-sided Student's t-test. **$p < 0.01$.

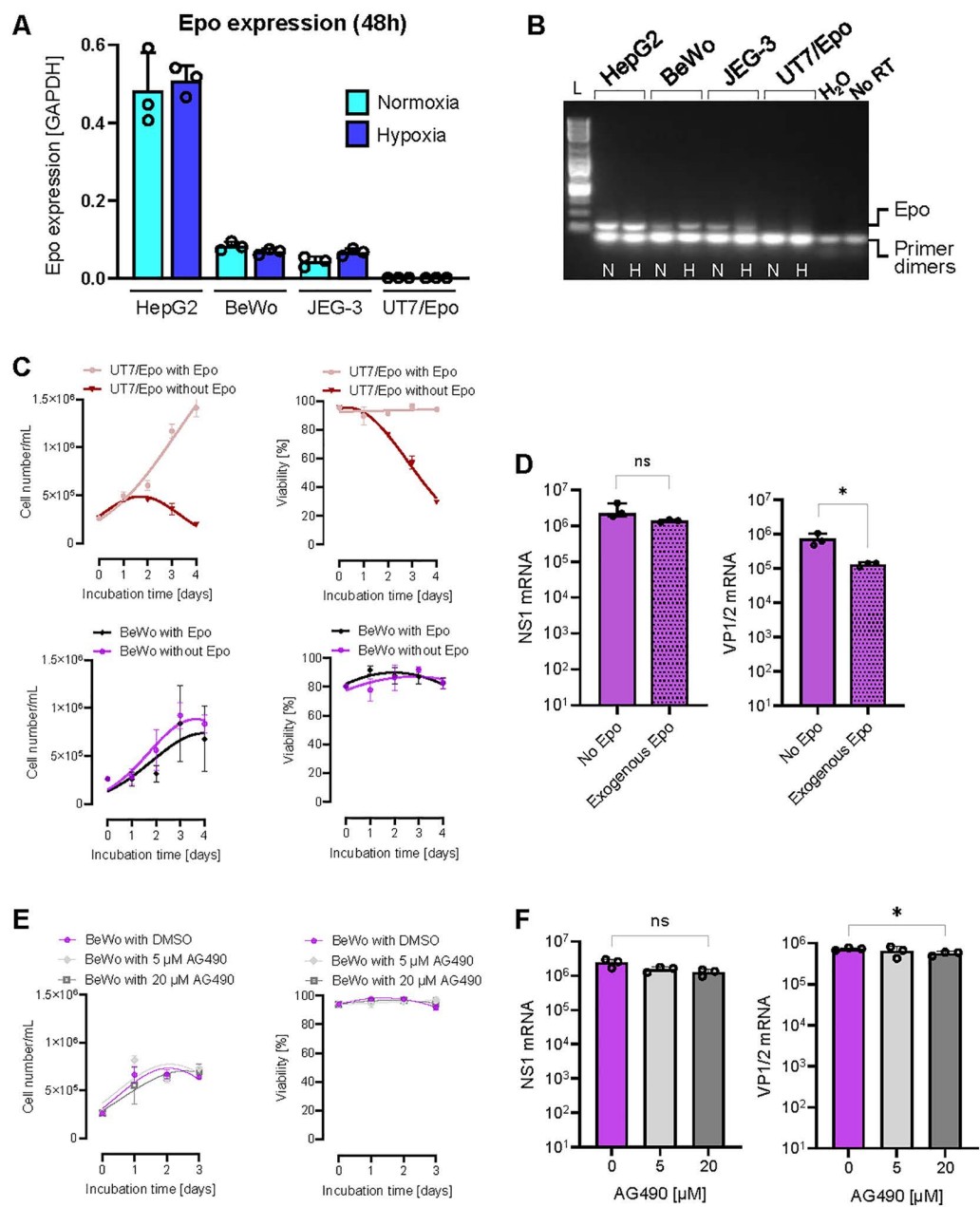

**Fig 8. EpoR signaling is dispensable for B19V infection in trophoblasts. (A)** Quantification of Epo mRNA levels in BeWo and JEG-3 cells under normoxia (N) and hypoxia **(H)**, with HepG2 and UT7/Epo as positive and negative controls, respectively. **(B)** Agarose gel electrophoresis of RT-PCR products showing expected amplicon size. L, 1 kb DNA ladder; no RT, no reverse transcriptase. **(C)** Effect of exogenous Epo supplementation on proliferation and viability of UT7/Epo and BeWo cells. **(D)** NS1 and VP mRNA levels in BeWo cells transfected with full-length B19V genomes, with or without Epo supplementation. **(E)** Cell proliferation and viability of BeWo cells treated with 5 or 20 µM AG490. **(F)** NS1 and VP mRNA expression in BeWo cells treated with AG490 prior to B19V genome transfection. Two-sided Student's t-test. *$p < 0.05$; ns, non-significant.

mRNA levels remained unchanged at both inhibitor concentrations, indicating that JAK2 activity does not influence viral gene expression in these cells (Fig 8F). Collectively, our findings indicate that B19V infection in trophoblasts does not rely on EpoR signaling, unlike its essential dependence in erythroid progenitor cells.

## Discussion

Human parvovirus B19 primarily infects EPCs in the bone marrow. While well known for causing hematopoietic dysfunction, B19V has gained increasing attention in the context of pregnancy due to its potential to induce severe fetal complications, including fetal anemia, non-immune fetal hydrops, and intrauterine fetal death [2]. These complications are more frequent when maternal infection occurs during the first half of pregnancy, coinciding with a developmental window characterized by heightened placental plasticity and rapid trophoblast proliferation [15–17]. Despite the recognition of these risks, knowledge of the mechanism of B19V transmission to the fetus remains poorly understood. As no targeted antiviral therapies or vaccines are currently available, the management of fetal B19V infection relies primarily on monitoring for signs of fetal anemia and, when necessary, performing intrauterine blood transfusions [53].

B19V entry is primarily mediated by VP1uR, a molecular entity or complex that has so far been detected exclusively in EPCs in the bone marrow [5–7]. This restricted expression profile underlies the pronounced erythroid tropism characteristic of B19V. In this study, we show that B19V internalizes into placental trophoblasts via the same receptor-mediated entry pathway, thereby expanding the known cellular tropism of the virus beyond the erythroid lineage. In contrast to CTBs and STBs, EVTs, which represent a terminally differentiated lineage derived from CTBs, do not support VP1u binding and, correspondingly, do not support B19V uptake and infection. This mirrors our observations during erythropoiesis, where VP1u uptake is observed in early erythroblasts but not in terminally differentiated erythroid cells [7]. Of note, the lack of detectable VP1u binding and uptake in the EVT model HTR/SVneo does not necessarily imply that placental EVTs lack VP1uR expression. This finding requires validation using additional EVT cell models and, ideally, confirmation in placental tissue.

Although the tyrosine kinase AXL has been proposed as a receptor for VP1u in erythroid cells [34], its broad expression across various human tissues [54] does not align with the strict erythroid tropism of B19V. Supporting this, our data show that EVTs, which express AXL but lack VP1uR, are not susceptible to B19V infection. In contrast, BeWo and JEG-3, which express VP1uR but not AXL, do support viral uptake. Together, these findings indicate that AXL is not required for B19V entry into placental trophoblasts.

Besides VP1uR, the glycosphingolipid globoside is required for the infection. Previous work demonstrated that B19V interacts with globoside only under acidic conditions [8], preventing unwanted interactions with globoside expressed in non-permissive cells. This pH-dependent interaction occurs in the acidic environment of the nasal mucosa, allowing the virus to attach to epithelial surfaces, undergo transcytosis, and be released at the basolateral side, where the neutral pH promotes dissociation from globoside [9]. Beyond the nasal mucosa, globoside is essential for endosomal escape of B19V following VP1uR-mediated endocytosis in EPCs. In globoside knockout UT7/Epo cells, incoming viruses remain trapped in endosomes, preventing infection. In these cells, infectivity is restored by PEI-induced endosomal rupture [10]. Numerous studies have shown that globoside is expressed in the human placenta [18,41,42,55], which we confirmed both in trophoblast cell lines and in cryosections from term placenta. Although first-trimester placenta sections were not available, these results strongly suggest that globoside is widely expressed in the human placenta. The only exception in our analyses was the choriocarcinoma-derived cell lines BeWo and JEG-3. The lack of globoside expression is likely due to the disrupted glycosylation pathways in these cancer-derived cell lines, which are known to exhibit altered glycosyltransferase expression compared to the normal cells from which they derive [43,44]. Although BeWo and JEG-3 cells express VP1uR and support viral internalization, productive replication is blocked, indicating that globoside is essential for B19V infection in trophoblasts. As in globoside KO UT7/Epo cells [10], PEI-mediated endosomal rupture restored viral replication, confirming that globoside is required for endosomal escape of B19V also in trophoblasts, underscoring this interaction as a potential antiviral target. These results also emphasize that, although BeWo and JEG-3 cells are widely used to study trophoblast biology, their lack of globoside expression renders them unsuitable for modeling B19V infection.

Our findings indicate that both CTB and STB populations support B19V entry. However, the markedly higher transcriptional activity observed in first-trimester CTBs, which are highly proliferative compared to the low-proliferating CTBs from term placenta suggests that cellular proliferation is a key determinant of infection. As trophoblasts transition toward a more

differentiated, functionally specialized state with reduced proliferative capacity in later gestation, their ability to support B19V replication appears to diminish. This supports a model in which the developmental stage of placental trophoblasts shapes their permissiveness to the infection and is consistent with the higher risk associated with maternal infection earlier in pregnancy. Supporting this, the first-trimester trophoblasts BeWo and JEG-3, whose high proliferation, further supported by their malignant nature, produce significantly higher levels of NS1 expression upon transfection than term placenta-derived hPTC^CTB cells. However, HTR-8/SVneo cells, which are EVTs from early placenta and exhibit similar proliferation rates to BeWo and JEG-3, generate lower NS1 mRNA levels. This suggests that, beyond proliferation, additional intrinsic intracellular factors contribute to the elevated permissiveness of early CTBs compared to EVTs.

Interestingly, EpoR signaling, which is required for B19V infection in erythroid cells through JAK2 activation [48], was dispensable in trophoblasts. Although the underlying mechanisms remain to be elucidated, this points to the existence of alternative, Epo-independent pathways that enable B19V gene expression in non-erythroid cells, thereby broadening our understanding of the cellular tropism and adaptability of B19V. Nevertheless, the lack of Epo dependence observed in trophoblast models does not exclude a role for Epo signaling in the context of the placenta.

Together, these observations suggest that B19V susceptibility in trophoblasts is governed by a combination of proliferative capacity and cell-intrinsic factors. Integrating prior observations on pH-dependent receptor usage [8–10,35] with the current results, we propose a model of B19V placental invasion involving sequential, pH-regulated receptor engagement. In this model, VP1uR mediates viral entry into non-proliferative syncytiotrophoblasts (STBs). Once internalized, the virus encounters the acidic environment of endosomes, leading to its dissociation from VP1uR and subsequent binding to globoside. This interaction facilitates viral transfer to the underlying CTBs, where conditions are favorable for infection.

In summary, this study demonstrates that B19V can enter placental trophoblasts, expanding its recognized strict tropism beyond the erythroid lineage to include early villous trophoblasts. The combined presence of VP1uR, globoside, and high proliferative capacity underlies the heightened susceptibility of the early placenta and positions first-trimester villous trophoblasts as the most relevant models for elucidating infection mechanisms and guiding the development of antiviral strategies to prevent vertical transmission.

## Materials and methods

### Ethics statement

Written informed consent was obtained from the patient. The study protocol was approved by the Ethics Committee of the Canton of Vaud, Switzerland (CER-VD 2022-00066) and complies with its ethical standards.

### Cells

UT7/Epo cells, provided by E. Morita (Tohoku University School of Medicine, Japan) were cultured in Eagle's Minimum Essential Medium (MEM, Thermo Fisher Scientific, Waltham, MA, USA) with 5% FCS and 2 U/mL Epo. UT7/Epo Gb4-KO cells were generated by co-transfecting a CRISPR/Cas9 GFP plasmid targeting β-1,3-Gal-T3 with an HDR RFP plasmid, followed by bulk and single-cell FACS sorting and confirmed by RT-qPCR as previously described [35]. The UT7/Epo globoside-KO cells are cultured in the same manner as UT7/Epo cells. The BeWo b30 Aberdeen choriocarcinoma cells were obtained from T. Bürki (Swiss Federal Laboratories for Material Science and Technology; EMPA, St. Gallen, Switzerland) and cultured in Ham's F-12K medium (Gibco, Waltham, MA, USA) containing 10% FCS and 50 U/mL of penicillin/streptomycin. BeWo b30 Aberdeen cells were differentiated into syncytiotrophoblasts with 10 μM of forskolin (72112, STEMCELL Technologies Switzerland GmbH, Basel, Switzerland) at 37°C for 72 h. The choriocarcinoma JEG-3 cells were obtained from C. Albrecht (Institute of Biochemistry and Molecular Medicine, University of Bern, Switzerland). The hepatocarcinoma cell line HepG2 was purchased from ATCC. JEG-3 and HepG2 cells were cultured in Eagle's MEM with 5% FCS and 50 U/mL penicillin/streptomycin. The extravillous trophoblast cell line HTR-8/SVneo was obtained from S. Rudloff (Department of Nephrology and Hypertension, University of Bern, Switzerland) and were reported to contain

a mix of trophoblasts and stromal cells [56]. The cells were grown in RPMI 1640 medium (Gibco) containing 5% FCS and 50 U/mL penicillin/streptomycin. The REH cells, a B-cell precursor leukaemia cell line, were cultured in RPMI 1640 medium containing 10% FCS and 50 U/mL penicillin/streptomycin. The aforementioned cells were routinely maintained under standard conditions at 37°C and 5% $CO_2$. hPTC$^{CTB}$, an immortalized human term placenta-derived cytotrophoblast cell line from the basal plate, was a gift from R. Menon (Department of Obstetrics and Gynecology, Maternal-Fetal Medicine, Perinatal Research Division, University of Texas). The cells were grown in DMEM:F12 medium supplemented with 0.2% FCS, 0.3% BSA, 1% amphotericin-B (AmpB), 1% ITS-X-supplement, 1% penicillin/streptomycin, 50 ng/mL EGF, 0.5 μM A83-01, 1 μM SB431542, 1.5 μg/mL L-ascorbic acid, 2 μM CHIR99021, 5 μM Y27632, 0.8 mM VPA, and 0.1 mM 2-mercaptoethanol and cultured at 37°C, 5% $CO_2$, and 5% $O_2$. The medium was replenished every second day. All cells were used at low passage numbers (P5–25) to minimize genetic drift and phenotypic changes.

## Cryosections of human placenta

Cryosections of human placenta were prepared from tissue obtained immediately after birth from a term pregnancy delivered by caesarean section. Full-thickness placental biopsies (~5 cm) were collected. Residual blood was removed by washing the biopsies with ice-cold PBS (Gibco) supplemented with 100 U/mL penicillin and 100 μg/mL streptomycin. The tissue was trimmed to approximately 1 cm³, embedded in 1% of low melting point agarose (Promega, Madison, WI, USA) in Peel-A-Way molds S-22 (Sigma, St. Louis, MO, USA) and kept on ice. Using a VT1200/S vibrating-blade microtome (Leica Microsystems), sections of 700 μm thickness were obtained with the following settings: speed 0.12-0.26 mm/s, amplitude 3 mm, angle 18°. The human placenta slices were transferred to 6 well plates (TPP) containing 3 mL DMEM GlutaMax (32430027, Gibco), 10% fetal bovine serum (10270, Gibco), 10 mM HEPES (1530056, Gibco), 1% Glutamine, 1% MEM-NEAA, 100 units/mL of penicillin and 100 μg/mL streptomycin. The cultures were maintained for 2 days at 37°C, 5% $CO_2$ prior infection and medium was changed every 24 h.

The specimens were fixed in 4% formalin (Formafix) over night at 4°C. For cryopreservation, the tissue was first incubated in 30% sucrose/PBS overnight, embedded in 7.5% gelatin/10% sucrose at 37°C for 45 min, and polymerized at room temperature. Samples were frozen in a dry ice/100% ethanol bath and stored at -80°C. Cryosections of 10 μm were cut using a Leica cryostat and mounted on Superfrost Plus glass slides (Thermo Fisher Scientific) as previously described [57].

## Viruses

Native B19V was obtained from de-identified plasma samples of infected, seronegative individuals, confirmed by virus-specific serology (CSL Behring, Bern, Switzerland). Infected plasma was thawed and clarified by centrifugation at 4,000 rpm for 10 minutes at 4°C. Viral genomes were quantified by quantitative PCR (qPCR) using Luna Universal One-Step Reaction Mix (M3003, New England Biolabs, Ipswich, MA, USA) with primers specific for the NS1-coding region as mentioned below. Plasmids containing the complete B19V genome were used as external standards in 10-fold serial dilutions.

## Expression and purification of recombinant VP1u and MS2 VLPs

*Escherichia coli* (*E. coli*) BL21(DE3) containing pT7-FLAG-MAT-Tag-2 vectors encoding for full-length or truncated (ΔN29, ΔC128) VP1u variants of B19V were grown in LB broth medium and induced with IPTG for protein expression. Proteins were harvested and purified by Ni-NTA affinity chromatography as described previously [5]. For click-chemistry, recombinant VP1u was reduced with 5 mM TCEP.

MS2-bacteriophage virus-like particles were expressed using *E. coli* BL21(DE3). MS2-VLPs were purified by ultracentrifugation through a 20% sucrose cushion as described elsewhere [7].

## Conjugation of VP1u to fluorescently labeled MS2 VLPs by click chemistry

MS2 VLPs were purified and labelled with NHS-Atto 488 (Atto-Tec, Siegen, Germany) using a 40-fold molar excess of dye. The reaction was quenched, and labeled VLPs were recovered by centrifugation through a 20% sucrose cushion to remove unreacted dye. Subsequently, MS2 VLPs were incubated with 500-fold molar excess of maleimide-PEG$_{24}$-NHS (22114, Thermo Fisher Scientific) for 1 h. Excess crosslinker was eliminated using a 40 kDa MWCO desalting column. The activated VLPs were then conjugated to reduced recombinant VP1u protein and further purified by a second 20% sucrose cushion [7].

## Drugs and reagents

Forskolin (STEMCELL Technologies Switzerland GmbH, Basel, Switzerland) was dissolved in DMSO at 10 mM. Linear 25 kDa polyethyleneimine (PEI; PolySciences, Warrington, PA, USA) was dissolved in water at 1 mM. A 3 kDa tetramethylrhodamine-labeled dextran (Invitrogen, Carlsbad, CA, USA) was dissolved in water at 20 mg/mL. CHIR99021 (SML1046), A83-01 (SML0788), SB431542 (616464), L-ascorbic acid (A4544), EGF (E4127), Penicillin/Streptomycin (P4333-100ML), and Amphotericin-B (A2942) were obtained from Sigma. CHIR99021 and A83-01 were dissolved in DMSO; L-ascorbic acid was dissolved in water; and EGF was dissolved in 0.85% NaCl. The reagent Y27632 was obtained from Millipore (6880001MG, Darmstadt, Germany) and dissolved in water. The ITS-X-supplement (51-500-056) and 2-mercaptoethanol (50-114-7851) were obtained from Gibco. BSA and the VPA were obtained from Thermo Fisher Scientific. AG490 (658401) was purchased from Sigma and dissolved in DMSO.

## Binding and internalization assays

Binding of recombinant VP1u (100 ng) and B19V (10$^5$ geq/cell) was performed at 4°C for 1 h in UT7/Epo, BeWo, JEG-3, hPTC$^{CTB}$, and REH cells resuspended in PBS (pH 7.2). Cells were washed with ice-cold PBS and then prepared for internalization, DNA extraction, or fixed for immunofluorescence. DNA was isolated using the GenCatch Plus Genomic DNA Miniprep Kit (1660250, Epoch Life Science, Missouri City, TX, USA). Internalization of recombinant VP1u (100 ng), B19V (10$^5$ geq/cell), and MS2-VP1u (10$^6$ geq/cell) was conducted in UT7/Epo, BeWo, JEG-3, hPTC$^{CTB}$, and REH cells at 37°C for 30 min. Cells were then washed and either fixed for immunofluorescence or processed for DNA extraction, as described above. Viral DNA was analysed by qPCR with the B19-NS1-F: 5′-GGGGCAGCATGTGTTAAG-3' and B19-NS1-R: 5′-CCATGCCATATACTGGAACAC-3′.

## Immunofluorescence

For surface detection of globoside, cells of interest were blocked with 1% bovine-serum albumin prior to incubation with a polyclonal chicken IgY anti-Gb4 antibody (JM07-164-4, J. Müthing, University of Münster, Germany) at 4°C for 30 minutes. Cells were washed multiple times with ice-cold PBS before fixation with 4% formaldehyde for 10 min, washed with PBS, quenched with 1 M Tris-HCl (pH 8.5) as previously described [9]. The antibody targeting globoside was detected with a polyclonal goat anti-chicken IgG antibody conjugated to Alexa Fluor 594 (ab150176, abcam, Cambridge, UK). Detection of VP1uR was assessed in cells of interest by incubating recombinant B19 VP1u at 37°C for 30 min in the presence of a monoclonal rabbit anti-FLAG IgG antibody (14793S, Cell Signaling Technologies, Danvers, MA, USA). The cells were washed several times with ice-cold PBS before fixation in a 1:1 mixture of methanol:acetone at -20°C for 4 min. The FLAG antibody was detected with a polyclonal Alexa Fluor 488 conjugated goat anti-rabbit IgG (A11008, Invitrogen). B19V was detected using a monoclonal mouse IgG anti-B19-VP2 antibody (521-5D; MAB8292, Millipore, Burlington, MA, USA) and a secondary polyclonal goat anti-mouse IgG conjugated to Alexa Fluor 488 (A11001, Invitrogen). The nonstructural protein NS1 was detected using a monoclonal human IgG anti-B19-NS1 antibody (1424; 94408, Mikrogen, Neuried, Germany) and a polyclonal goat anti-human IgG antibody conjugated to Alexa Fluor 488 (A11013, Invitrogen). AXL was detected

using a polyclonal rabbit IgG anti-AXL antibody (13196-1-AP, Proteintech, Manchester, UK) and a polyclonal goat anti-rabbit IgG antibody conjugated to Alexa Fluor 488 (A11008; Invitrogen). The samples were mounted with EMS shield mount (17985-09, Electron Microscopy Sciences, Hatfield, PA, USA) containing 0.2 ng/mL DAPI. The samples were visualized with a 63 × oil immersion objective by laser scanning confocal microscopy (LSM880, Zeiss, Oberkochen, Germany).

For immunofluorescence analysis of FFPE slices, gelatin was removed, formalin was quenched with 1 M Tris-HCl, pH 8, tissue was permeabilized with 0.2% Triton-X-100 and blocked with 5% milk/PBS. Cytotrophoblasts were labeled using a polyclonal goat IgG anti-TROP-2 antibody (1:100, PA5–47030, Thermo Fisher Scientific) followed by a polyclonal chicken anti-goat IgG antibody conjugated to Alexa Fluor 594 (A21468, Thermo Fisher Scientific). Syncytiotrophoblasts were detected using a monoclonal mouse IgG anti-Bcl-2 antibody (1:50, MCA1550, Bio-Rad, Hercules, CA, USA), and the apical surface of syncytiotrophoblasts was detected using a monoclonal mouse IgG anti-Syndecan-1 antibody (1:100, A279775, Antibodies.com, Cambridge, UK) and a polyclonal goat anti-mouse IgG antibody conjugated to Alexa Fluor 488 (A11001, Thermo Fisher Scientific). All antibodies were diluted in 2% milk/PBS and carried out at 4°C, after which specimens were treated with 2 mM $CuSO_4$/50 mM $NH_4$. Specimen was mounted with EMS shield mount as described above and visualized using a 63 × oil immersion objective by LSM880.

### Infectivity assays

Cells were infected with B19V ($10^5$ geq/cell) at 37°C for 30 min before cells were washed with PBS and then either incubated further at 37°C or processed for RNA extraction using the GenCatch Total RNA Miniprep Kit (1660250, Epoch Life Science). Extracted RNA was treated with 1 U/µl RQ1 DNase I (M610A, Promega) at 37°C for 1 h before NS1 mRNA was quantified by RT-qPCR using the primers as previously specified. When PEI was used for infection, cells were inoculated with PEI (15 µM) simultaneously and washed with PBS at specific times post-infection. RNA was extracted and analyzed by RT-qPCR. Following primers were used: NS1-F: 5′-GGGGCAGCATGTGTTAAG-3′, B19-NS1-R: 5′-CCATGCCATATACTG GAACAC -3′, VP1/2-F: 5′-CATGCACACCTACTTTCCCAA-3′, and VP1/2-R: 5′-TGATGCAAACCCCATCCTCC-3′.

Transfection of B19V DNA was carried out with Lipofectamine 3000 Transfection Reagent (L3000001, Invitrogen) following the manufacturer's protocol closely.

### Endosomal integrity

To assess the effect of PEI on endosomal integrity, cells were incubated with increasing concentrations of PEI and low-molecular weight tetramethylrhodamine-labeled dextran (3 kDa) at 37°C for 30 min and washed several times with PBS. Live-cell images were taken using the LSM880 microscope. Fluorescence intensity was measured in 50 cells per condition using ImageJ (National Institutes of Health, Bethesda, MD, USA).

### Transfection

For transfection, cells between passages 5–25 were seeded in 24-well plates using standard growth medium and transfected the following day with 100 ng of purified B19V DNA. Transfection was carried out with Lipofectamine 3000 Transfection Reagent (L3000001, Invitrogen) according to the manufacturer's protocol. Two days post-transfection, cells were washed several times with PBS and DNA was extracted with Epoch Kit as described above.

To analyse the transfection efficiency, cells were transfected with 100 ng GFP-expressing plasmid DNA (Lonza, Basel, Switzerland) using Lipofectamine 3000. The nuclei were stained with Hoechst (1 µg/mL) at 37°C for 10 min for live-cell imaging. Images were captured by LSM880 confocal microscopy. The proportion of transfected cells was quantified using ImageJ by counting GFP-positive cells relative to the total number of Hoechst-stained nuclei. Therefore, channels corresponding to DAPI and GFP were separated, and the intensity thresholds were adjusted to a range of 100–255. Cell segmentation was performed by converting images to binary format and touching objects were separated. Particle

analysis was conducted with a size range of 0.0005 cm² to infinity and circularity set from 0.00 to 1.00. The number of GFP-positive (transfected) and DAPI-positive (total) cells was quantified. Transfection efficiency was calculated as the percentage of GFP-positive cells relative to the total number of cells.

## Cell proliferation assays

Cell proliferation was monitored over four days. Cells were seeded in standard growth medium in 24-well plates and incubated for zero to four days. At each timepoint, cells were washed with PBS, lysed in the well and genomic DNA was extracted using the Epoch DNA Extraction Kit as mentioned above. The β-actin gene was quantified by qPCR using the following primers: β-Actin-F: 5'-CCACCATGTACC CTGGCATT-3', β-Actin-R: 5'-CGGACTCGTCATACTCCTGC-3' and displayed as the percentual increase over day 0.

Cell number and viability were monitored over 3–4 days. Cells were either left untreated or treated with 2 U/mL Epo or 20 µM AG490. At each time point, adherent cells were detached using Accutase (Sigma), resuspended in standard growth medium to the desired final volume, and mixed 1:1 (v/v) with 0.2% trypan blue. Total cell counts and viability were then measured using a LUNA automated cell counter (Logos Biosystems, Anyang, South Korea).

## Detection of glycosyltransferase and Epo mRNA expression

The expression of glycosyltransferase mRNAs involved in the synthesis of Gb3 and Gb4 was assessed by extracting total RNA from the cells of interest followed by DNase-I treatment. The mRNA of the 4-α-galactosyltransferase (A4GalT) was quantified by RT-qPCR using the primers A4GalT-F: 5'-GCCTCCAGGATCGCACTCAT-3' and A4GalT-R: 5'-CATGCACAG CGCCATGAACT-3'. This enzyme adds a galactose to lactosylceramide, creating trihexosylceramide (Gb3). The mRNA of the 3-β-N-acetylgalactosaminyltransferase (β3GalNT1) was quantified using the primers β3GalNT1-F: 5'-CTCCT GAGTTTCTTTGTG ATGTGG-3' and β3GalNT1-R: 5'-CATTACGTACTTGGCATTGGGG-3'. This enzyme initiates the final enzymatic step that adds the N-acetylgalactosamine to Gb3, producing Gb4. GAPDH mRNA quantification was used for data normalization using the following primers: GAPDH-F: 5'-GCCAAAAGGGTCATCATCTCTG-3' and GAPDH-R: 5'-CCTGCTTCACCACCTTCTTG-3'.

The expression of endogenous Epo mRNA was quantified by RT-qPCR using the primers Epo-F: 5'-CAGCTGCATGT GGATAAAGCC-3' and Epo-R: 5'- ATTGTTCGGAGTGGAGCAGC-3'. Data are expressed as relative mRNA levels normalized to GAPDH, rather than as absolute transcript copy numbers per cell.

## Quantification and statistical analysis

Statistical analyses were carried out using GraphPad Prism version 10.3.1 (GraphPad Software, Boston, Massachusetts USA). Details about the statistical method used are mentioned in the legend section of the respective figures. Error bars indicate SD as mentioned in the respective figure legends. In all cases, a p value $<0.05$ is considered significant.

## Supporting information

**S1 Fig. Expression of AXL in trophoblasts does not correlate with VP1uR expression and B19V uptake. (A)** Relative expression of AXL mRNA in UT7/Epo and in different trophoblast subtypes. mRNA levels were measured by RT-qPCR and normalized to GAPDH mRNA. **(B)** RT-PCR amplicons from BeWo and UT7/Epo cells were visualized by agarose gel electrophoresis. GAPDH, loading control; No RT, no reverse transcriptase; L, 1 kb DNA ladder. **(C)** Immunostaining of UT7/Epo and trophoblasts with an anti-AXL antibody (green). DAPI (Blue). Scale bar, 20 µm. All results are presented as the mean ± SD of three independent experiments.
(TIF)

**S2 Fig. Lack of globoside synthase (Gb4) expression in BeWo cells under various conditions. (A)** Relative mRNA expression of Gb3 and Gb4 in BeWo cells treated or untreated with forskolin (FSK), normalized to GAPDH mRNA, to

assess the effect of syncytiotrophoblast differentiation on Gb3/Gb4 expression. **(B)** Relative mRNA expression of Gb3 and Gb4 in BeWo cells exposed to normoxia, normoxia plus Epo, hypoxia, or hypoxia plus Epo, normalized to GAPDH mRNA, to assess the effect of oxygen concentration and Epo on Gb3/Gb4 expression. **(C)** Agarose gel electrophoresis of RT-PCR products corresponding to Gb3, Gb4, and GAPDH in BeWo cells under the conditions described in **(B)**, and in UT7/Epo cells incubated with or without Epo under normoxia. Each panel displays the PCR products for Gb3, Gb4, or GAPDH as indicated. No RT, no reverse transcriptase; 1 kb DNA ladder. All results are presented as the mean ± SD of three independent experiments. Statistical significance was calculated using two-sided Student's t-test. **$p < 0.05$; ns, non-significant.
(TIF)

**S3 Fig. Expression of globoside in term placenta. (A)** Trophoblast markers BCL-2, Trop-2 (both in green), and CD138 (red), detected in term placenta cryosections. DAPI (Blue). Scale bar, 40 μm. **(B)** Globoside (red), detected in term placenta cryosections. DAPI (Blue). Scale bar, 40 μm.
(TIF)

**S4 Fig. Time-dependent increase in β-actin mRNA levels as a proxy for cell growth in trophoblast subtypes.** Relative β-actin DNA levels were quantified by RT-qPCR in different trophoblast subtypes, UT7/Epo and REH cells, at days 1, 2, 3, and 4 of culture. Data are expressed as a percentage of β-actin expression relative to day 0. All results are presented as the mean ± SD of three independent experiments.
(TIF)

**S1 Data. Tab A**: Fig 3A. Relative expression of Gb3 and Gb4 mRNA in different trophoblast subtypes normalized to GAPDH mRNA. Fig 3C. Relative expression of Gb3 and Gb4 mRNA in hPTCCTB cells with and without Forskolin differentiation into STBs. **Tab B**: Fig 4A. B19V uptake in trophoblasts, UT7/Epo, and REH cells with or without trypsin treatment. **Tab C**: Fig 5A. Quantification of NS1 mRNA after B19V infection in different trophoblast subtypes and UT7/Epo cells. Fig 5B. Quantification of VP1/2 mRNA after B19V infection in different trophoblast subtypes and UT7/Epo cells. **Tab D**: Fig 6A. Transfection efficiency with a plasmid encoding green fluorescent protein (GFP), GFP positive cells were quantified as percentage of total cells. Fig 6B. NS1 mRNA expression in trophoblasts and UT7/Epo cells after B19V genome transfection. Fig 6C. VP1/2 mRNA expression in trophoblasts and UT7/Epo cells after B19V genome transfection. **Tab E**: Fig 7B. Quantification of cells with cytosolic distribution of dextrans in BeWo and JEG-3 cells in response to PEI treatment. Fig 7C. Quantification of NS1 mRNA levels at 24 h post-infection in BeWo cells treated or untreated with PEI. Fig 7E. Quantification of structural VP1/VP2 mRNA levels at 24 h and 48 h post-infection in BeWo cells treated or untreated with 5 μM PEI. **Tab F**: Fig 8A. Quantification of Epo mRNA levels in BeWo and JEG-3 cells under normoxia (Nx) and hypoxia (Hx). Fig 8C. Effect of exogenous Epo supplementation on proliferation and viability of UT7/Epo and BeWo cells. Fig 8D. NS1 and VP mRNA levels in BeWo cells transfected with full-length B19V genomes, with or without Epo supplementation. Fig 8E. Cell proliferation and viability of BeWo cells treated with 5 or 20 μM AG490. Fig 8F. NS1 and VP1/2 mRNA expression in BeWo cells treated with AG490 prior to B19V genome transfection. **Tab G**: S1A Fig. Relative expression of AXL mRNA in UT7/Epo and in different trophoblasts, normalized to GAPDH. **Tab H**: S2A Fig. Relative mRNA expression of Gb3 and Gb4 in BeWo cells treated or untreated with forskolin (FSK) normalized to GAPDH. S2B Fig. Relative mRNA expression of Gb3 and Gb4 in BeWo cells exposed to normoxia or hypoxia with or without Epo, normalized to GAPDH. **Tab I**: S4 Fig. Time-dependent increase in β-actin mRNA levels as a proxy for cell growth in trophoblasts, expressed as % of β-actin expression relative to day 0.
(XLSX)

## Acknowledgments

We thank Tina Bürki (Swiss Federal Laboratories for Material Science and Technology, EMPA) for kindly providing BeWo b30 Aberdeen choriocarcinoma cells. We are grateful to Christiane Albrecht and Stefan Rudloff (University of Bern) for providing JEG-3 and HTR-8/SVneo cells, respectively, and Ramkumar Menon (University of Texas) for the hPTCCTB cells.

## Author contributions

**Conceptualization:** Corinne Suter, Carlos Ros.

**Formal analysis:** Corinne Suter, Melanie Küffer, Carlos Ros.

**Funding acquisition:** Carlos Ros.

**Investigation:** Corinne Suter, Melanie Küffer, Amal Fahmi.

**Methodology:** Corinne Suter, Jan Bieri.

**Project administration:** Carlos Ros.

**Resources:** Corinne Suter, Amal Fahmi, David Baud, Marco P. Alves, Carlos Ros.

**Supervision:** Carlos Ros.

**Validation:** Corinne Suter.

**Visualization:** Corinne Suter.

**Writing – original draft:** Corinne Suter, Carlos Ros.

**Writing – review & editing:** Jan Bieri, Marco P. Alves.

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
