## [Decision Letter · Decision Letter 0]

20 Nov 2025

Cellular determinants of parvovirus B19 infection in the human placenta

PLOS Pathogens

Dear Dr. Ros,

Thank you for submitting your manuscript to PLOS Pathogens. After careful consideration, we feel that it has merit but does not fully meet PLOS Pathogens's publication criteria as it currently stands. Therefore, we invite you to submit a revised version of the manuscript that addresses the points raised during the review process. Please pay close attention to the suggestions for additional experiments made by Reviewers 1 and 2 (described below). Reviewer 1 recommended experimental validation of viral DNA and viral proteins at extended timepoints to corroborate your mRNA data and testing for progeny viral particles produced by the target cells. Reviewer 2 requested additional negative controls for the RT-qPCR data presented in the manuscript. Both R1 and R2 had suggestions for improving the clarity and honing the overall messages of Figures 5 and 6. Additionally, all three reviewers have made extensive suggestions on on improving the specificity of the language which will enhance the clarity of your manuscript.

We look forward to receiving your revised manuscript.

Kind regards,

Kinjal Majumder, PhD

Guest Editor

PLOS Pathogens

Donna Neumann

Section Editor

Editor-in-Chief

PLOS Pathogens

PLOS Pathogens

orcid.org/0000-0002-7699-2064

**Journal Requirements:**

At this stage, the following Authors/Authors require contributions: Corinne Suter, Melanie Küffer, Jan Bieri, Amal Fahmi, David Baud, Marco Alves, and Carlos Ros. Please ensure that the full contributions of each author are acknowledged in the "Add/Edit/Remove Authors" section of our submission form.

- ® on pages: 25, and 26

- TM on pages: 27, and 28.

5) We notice that your supplementary Figures are included in the manuscript file. Please remove them and upload them with the file type 'Supporting Information'. Please ensure that each Supporting Information file has a legend listed in the manuscript after the references list.

Potential Copyright Issues:

i) Figures 1A, and 2A. Please confirm whether you drew the images / clip-art within the figure panels by hand. If you did not draw the images, please provide (a) a link to the source of the images or icons and their license / terms of use; or (b) written permission from the copyright holder to publish the images or icons under our CC BY 4.0 license. Alternatively, you may replace the images with open source alternatives. See these open source resources you may use to replace images / clip-art:

7) We note that your Data Availability Statement is currently as follows: "All relevant data are within the manuscript and its Supporting Information files". Please confirm at this time whether or not your submission contains all raw data required to replicate the results of your study. Authors must share the “minimal data set” for their submission. PLOS defines the minimal data set to consist of the data required to replicate all study findings reported in the article, as well as related metadata and methods (https://journals.plos.org/plosone/s/data-availability#loc-minimal-data-set-definition).

**Reviewers' Comments:**

Reviewer's Responses to Questions

**Part I - Summary**

Reviewer #1: In the manuscript entitled " Cellular determinants of parvovirus B19 infection in the human placenta", Suter and collogues employ a wide range of sophisticated techniques developed by the Ros laboratory over the years, to address the issue of B19V ability to infect different placenta cell types.

The topic is of great interest, given the ability of B19V to cause severe disease and even death of the fetus in cases of congenital infection. Given the absence of approved, safe, and efficient antivirals, B19V congenital infection represents a major medical problem. Since B19V is believed to productively infect exclusively EPCs, it is not clear how fetal infection would occur, and the ability of the virus to infect the placenta is still unknown.

To address this issue the authors first assess the ability of VP1 (the viral attachment protein) to bind to the cell surface of several cell types of placental origin. For certain cell types the authors clearly show efficient binding, that suggests the ability to interact with cell surface constituents. Secondly, the ability of VP1 expressing VLPs to be internalized is shown, further supports the possibility that B19V might interact with placental cell types. Subsequently, the authors show that after infection or transfection certain cell types allow expression of structural protein mRNA.

Based on such evidences, the authors conclude that specific placental cell types express the still-unidentified Vp1 receptor (VP1uR) and are permissive to B19V replication.

While the study addresses an interesting question and data is of high quality, its interpretation suffers from two major ambiguities that, in my view, significantly affect the strength and balance of the conclusions.

First, the authors frequently associate the ability of certain cell types to bind VP1-containing particles with the presence of a putative VP1 receptor (VP1uR). Given that VP1uR remains unidentified at the molecular level, this interpretation is speculative and should be presented with greater caution. The observed binding or uptake could reflect non-specific interactions or alternative entry mechanisms. I do not think it is possible to prove the expression of an unidentified factor.

Second, the authors equate detection of VP1 mRNA with cellular permissiveness to B19V infection. Expression of viral transcripts alone does not demonstrate productive infection, which can only be established by showing viral DNA replication and production of infectious progeny.

Reviewer #2: This paper is a very interesting and important piece of work and a nice follow-up of their excellent previous work on the parvovirus B19 pH-dependent receptor switch and cell entry. This study shows an immense workload with many angles and different kinds of experiments to prove that B19V targets placental trophoblasts and uses a VP1u-specific (yet unknown) receptor, VP1uR, to enter placenta-derived cell lines and another receptor, globoside, to be released from the endosomes, leading to nuclear expression of its genes. They also show which placental-derived cells have either or both of these two receptors and which do not. This paper brings novel insights on how B19V infects the placenta in pregnancy. However, there are a few issues that will need clarification, outlined in Parts II and III below.

Reviewer #3: In this manuscript by Suter et al. entitled “Cellular determinants of parvovirus B19 infection in the human placenta”, the authors analyze in detail the susceptibility of three trophoblast cell types to infection by human parvovirus B19. The viral cellular entry and infection in cells were analyzed using detection of B19V receptor, binding and internalization assays, combined with confocal microscopy. The results show that trophoblast cells from the maternal-fetal interface, located between the fetal placenta and the maternal uterus, enable B19V infection. These findings expand the tropism of B19V to include trophoblast cells, thereby increasing our understanding of how the virus can cross the placenta and infect the fetus. The data are novel and informative, and the conclusion will be of interest to the virology, medical, and cell biology communities, as well as a broad audience.

**Part II – Major Issues: Key Experiments Required for Acceptance**

Reviewer #1: The interpretative issues mentioned above introduce an element of overstatement in the manuscript’s narrative. While the experimental observations are interesting, (I) the conclusions regarding receptor usage should be substantially tempered to ensure they are consistent with the data presented. Secondly (II), permissiveness of placental cell types should be experimentally demonstrated by showing an increase over time of viral DNA, and by the production of viral progeny. Finally (III) attempts to characterize viral entry in placental cell types should be performed.

Specific comments

• Figure 5. The data shown is interesting, however the authors evaluated B19V gene expression exclusively at the mRNA level and only at 2, 24 and 48 h post-infection. I strongly recommend performing immunofluorescence (IF) or Western blotting analyses to confirm protein expression of key viral markers (e.g., VP1/VP2, NS1) expanding the time line to 120 h. In parallel, quantification of viral genomic DNA at both 48 h and 120 h post-infection is essential to assess viral replication. Demonstration of viral progeny production should be attempted. Further the entry mechanism has not been characterized here. The authors are world leaders in B19V entry and should attempt at least some initial characterization to compare the entry pathway between placental cell types and UT7E cells, as shown in (PMID: 22718826).

• Figure 6. The data presented are not fully convincing. The authors evaluate B19V gene expression exclusively at the mRNA level and only at 48 h post-infection. While these results indicate transcriptional activity, they do not demonstrate productive viral replication. I strongly recommend performing immunofluorescence (IF) or Western blotting analyses to confirm protein expression of key viral markers (e.g., VP1/VP2, NS1). In addition, quantification of viral genomic DNA at both 48 h and 120 h post-infection is essential to assess viral replication dynamics. A productive infection should be reflected by a clear increase in viral DNA copy number over time. Finally, to conclusively demonstrate productive infection, the authors should measure intracellular and extracellular infectious viral titers, for example by testing infectivity in UT7/EpoS1 cells. This point is particularly critical, as the claim that these cells are permissive for B19V infection represents arguably the main finding of the manuscript. If confirmed, it would have important implications by suggesting that B19V may productively infect cell types outside the erythroid progenitor lineage. Robust molecular and functional evidence is therefore required to substantiate this conclusion.

major points:

Figures 5 and 6

The authors should extend their analysis to include later time points post-infection or post-transfection (e.g., 120 h p.i./p.t.) to better assess the kinetics of viral replication and gene expression. Specifically, they should:

• Evaluate the production of viral antigens (NS1, VP1/VP2) by Western blotting and/or immunofluorescence (IF) to confirm protein-level expression.

• Quantify viral genome copy number over time to determine whether viral DNA replication occurs and increases at later stages.

• Assess productive infection by titrating infectious viral particles in UT7/EpoS1 cells, using both cell lysates and culture supernatants collected at multiple time points post-infection.

Figure 5

The authors should also further characterize the viral entry pathway. It would be important to clarify whether the same entry mechanism operates in placental cells and in UT7/EpoS1 cells. Sensitivity to endosomal acidification inhibitors—such as Bafilomycin A1, ammonium chloride, or chloroquine—should be tested to determine whether B19V entry is pH-dependent in both cell types.

These additional experiments would substantially strengthen the mechanistic conclusions and clarify whether the observed infection is truly productive and comparable across different cell systems.

Reviewer #2: 1. RT-PCR of B19V NS1 and VP1/2 mRNAs: Please provide evidence that the RT-PCR products are truly from viral mRNA, and not from carry-over DNA. The DNase treatment for 1 hour does not always destroy all DNA, so a no-RT control should always be included to notice DNA carry over. Should further be mentioned whether or not these target regions were chosen to span splice sites, which is preferable, and if so, the amplicons should be sequenced (or by gel electrophoresis) to show the shorter length of mRNA-derived cDNA to that of genomic DNA. Please add at least some such evidence information to the methods section and discuss it in Discussion. Also, give manufacturer and concentration of the DNase enzyme used, there are big differences in efficiency. It is important to prove that the RT-PCR results are not from DNA, a common mistake in publications.

Especially in Fig 5, it seems that all cells (including REH) express viral mRNAs, up to 10E3 copies, even at 1 h pi, at which point it is supposed to have only the input virus. Line 267: “VP1/2 transcripts were undetectable” is thereby not correct. Moreover, all non-permissive cells (except REH), especially BeWo and HTR-8/SVneo cells, showed statistically more NS1 mRNA expression at 24 and 48 h than at 1 h, with ∗∗∗p < 0.001, which is even more significant increase than for the permissive UT7/Epo cells (at ∗∗p < 0.01). Where do these mRNAs come from, if the virus can neither internalize (HTR-8/SVneo and REH cells) nor reach the nucleus (BeWo, JEG-3)? Do the mRNA loads go hand in hand with the genome loads? Also, how come the signal is increasing with time in the non-permissive cells? If input has mRNA, it suggests the RT-PCR indeed shows carry-over DNA.

Reviewer #3: The transfection of B19V genomes effectively eliminates the role of entry pathway-induced variation of infection. However, additional intrinsic intracellular factors affecting the progression of infection should be discussed in more detail in the discussion. Knowledge of potential intracellular bottlenecks is available in the literature.

Authors refer to VP1uR expression in the cell lines based on an indirect method of VP1uR recognition through the internalization of its counterpart VP1u into the cells without analyzing actual VP1uR expression levels in the cells. Please rephrase the title and text accordingly throughout the manuscript.

**Part III – Minor Issues: Editorial and Data Presentation Modifications**

Reviewer #1: minor points

1. line 145 " These FLAG-tagged VP1u constructs were incubated with ..." I don't think incubation involved constructs (usually refers to plasmid DNA):

2. line 154 " These findings reveal, for the first time, the expression of the B19V-specific receptor VP1uR beyond the erythroid lineage." I am not sure these findings actually reveal what the authors are trying to imply. Which receptor is actually used in unclear at the moment.

3. Line 170 " These results confirm the presence of functional VP1uR in CTBs from various placental origins and its absence in EVTs." Again, I am not convinced of the authors conclusion, The presence of VP1uR is not formally demonstrated. The identity of VP1uR is currently unknown, and therefore it is impossible to demonstrate its presence!

4. Line 253 " To determine whether B19V internalization in trophoblast cells results in productive infection, the expression of the early viral transcript NS1 was analyzed 24 h and 48 h post-infection." Detection of viral transcripts does not mean there is productive infection. To demonstrate a productive infection viral DNA must be replicated and viral particles produced.

5. Line 301 " Notably, the substantial increases in NS1 and VP1/VP2 mRNA expression observed in JEG-3 and BeWo cells far exceeded the differences in transfection efficiency, underscoring the higher permissiveness of these CTB-derived cell lines to B19V replication." Again, I see no proof of B19V replication.

6. Line 307 " Overall, the findings indicate that the intracellular environment of first-trimester CTBs is significantly more permissive to B19V replication than that of EVTs or CTBs derived from term placentas." Permissiveness is not assessed.

Reviewer #2: 1. Please write out the abbreviations first time mentioned, very tiring to have to search the last section (Materials and Methods) for explanations, and not always finding it even there.

2. Lines 39 and 81: VP1uR, the erythroid-specific receptor” and “The VP1uR has been exclusively found in the target EPCs in the bone marrow, explaining the marked erythroid cell tropism of the virus”. How has this unknown molecule been found on EPCs and is it really specific or exclusive? In the references given, binding was searched for in only a few cell types, but not all human cells, so how can it be stated that it truly would be specific for erythroid cells? How about fetal myocardial cells, endothelial cells or liver cells? This current manuscript is actually one piece of evidence against it being specific, and there can be other cell types. Despite the heroic efforts to exclude globoside and AXL as being true cell receptors in the studied malignant cells (and earlier EPCs), there may still be a role in some cells of other known or unknown receptors, including the antibody-enhanced entry. Exact and correct wording is important so the reader would know what really are proven facts and what are not quite yet, even though perhaps likely. Perhaps using words like “has hitherto been” or “so far” or “it seems to” or similar, depending on the context and facts. The “highly restricted” on line 133 is also better in this regard.

3. Line 96: Surely there are newer reviews than ref 12 from 1997, or original studies?

4. Line 99: perhaps use the word “risk” instead of “chance” of adverse effect.

5. Replication per se is actually not shown, as opposed to only VP1u or virion entry, or VP protein by IF after binding or mRNA expression after transfection of B19V genomes in the various cell cultures. Viral genomes were quantified by qPCR only for Fig 4, and this was after 1 hour incubation, which is not yet replication (Lines 551-552). Mere mRNA expression is not proof of replication. If no replication experiments can be done e.g., in normal placental cells, (which would increase the impact of this paper), at least modify the wordings. Both words “infection” and “replication” are mentioned in many places, of which some do not seem to be correct. Moreover, “permissiveness” refers to a cell's ability to support a virus' complete replication cycle, not only mRNA expression after transfection. Please check that these words really mean what they should in the different contexts (including section and figure headings); is it transfection or infection and expression/transcription or replication…

6. Lines 146-150: The cells used are all malignant immortal cell lines that may have extended viral tropisms and modified transcriptomes, which the lack of globoside proves (as said on lines 232-234 and 323-324 and 437-440 and lines 446-448). How do the authors know what is similar to the real normal placental cells? Further, Abou-Kheir et al. (Placenta 50; 2017) point out that the HTR-8/SVneo cell line contains a mixed population of cells. Further, the hPTCCTB cells are, contrary to the other cells, derived from the placental basal plate. A more detailed description of these cell lines should be included either here or in Materials and Methods, and discussed in Discussion.

The authors did obtain also normal placenta, and even cultured those cells; please mention why these were not infected with B19V (or VP1u constructs) and included in these B19V experiments (only globoside was now studied in these cells that is already known to be expressed)?

7. Line 155 says “B19V-specific receptor VP1uR”. How do the authors know this receptor is “B19V specific”? VP1uR may very well bind also other pathogens, as do globoside. Please clarify and re-write.

8. Lines 190, 700-710, 754 and 759: In the RT-PCR for globoside Gb3 and Gb4 mRNAs, what were the mRNA levels? It is only told (actually twice in the Fig 3 figure legend) that “mRNA levels were measured by RT-qPCR and normalized to GAPDH expression”. However, it is not revealed how GAPDH relates to the cell amount, i.e., how to interpret the Y axis of Fig 3, does it show the Gb mRNA copies per cell or per 1000 cells or a million cells, or what? Same for the viral mRNAs.

Line 708: Is this GAPDH quantification of mRNA or DNA? qPCR would mean DNA. Did you have a positive control for mRNA? Please use italics for gene names to differentiate mRNA from the actual protein product (for NS1, VP1/2, Gb, GAPDH and Epo)

9. Lines 239-240 and Fig 4: Does trypsin really remove the B19V capsids? Are there proof of this? Does trypsin cleave either the capsid or the VP1uR? In Fig 4 the cells look the same with and without trypsinization. Please explain.

10. Line 245-249 and Fig 4: “A strong viral signal was observed in BeWo cells, weaker staining in JEG-3 and hPTCCTB, and no detectable signal in HTR-8/SVneo”. However, this is not correct; in Fig 4B there is no signal in hPTCCTB cells either, please correct.

Line 248: Fig 1 does not really show “levels of VP1uR expression” (there is no RT-PCR for VP1uR), but instead it shows binding of VP1u, which should correlate with the presence of its receptor. This wording may confuse the reader, so best to be specific and exact with the wording.

11. Line 272: How many % of cells were “infected” in each cell culture, did the yield correspond to the % infected? This would need to be calculated, instead of normalising to GAPDH expression that occurs in all cells.

12. Line 275 and 346: What is “structural mRNA expression”? Please correct the wordings.

13. Lines 284-315 and Fig 6: Transfection efficiencies were quite low and differed between cell types. The expression of e.g., hPTCctb could proportionately be much higher if taking into account that only a small fraction of cells was transfected. Same stands for the infection efficiencies in the various cells (see comment 11). Were the expression levels calculated for the proportion of transfected (or infected) cells or all cells? If the latter, there is a bias depending on the fraction of cells that contain the virus genome, so this should be calculated, or at least discussed.

On line 302-304 this subject is approached, but it is not really understood, and the math is lacking: The B19V non-permissive JEG-3 and BeWo cells have both higher transfection efficiencies and higher viral mRNA expression efficiencies than the B19V-permissive UT7/Epo cells, which proves my point, the actual transfected cells may have high transcription rate but the overall transcription is still low because too few cells have been transfected.

Furthermore, the JEG-3 and BeWo cells cannot be described as B19V permissive, as now is indicated on line 303?? However, another statement later (lines 314-315) summarizes that these cells are indeed resistant to B19V infection, but the reader may be confused if the wording is not carefully chosen…] The heading of Figure 6 is thus also misleading in two places: “Permissiveness of trophoblast subtypes to B19V infection.” (see comment 3 above) Please correct.

14. Line 326: Polyethyleneimine is misspelled.

15. Line 360: How was the Epo mRNA quantified?

16. Lines 371-372 and 466: This interpretation is perhaps not the full truth. This may also suggest that BeWo cells do not need Epo due to the malignant transformation and rapid growth, but perhaps normal trophoblasts, that truly are dependent on Epo (like UT7/Epo cells are), would need Epo also to permit B19V replication? Difficult to use malignant BeWo cells for studying normal placenta behaviour (see comment 6 above).

17. Line 394: This sentence should be “These complications are more frequent when infection of the mother occurs…” Those references, and others, say that the highest risk for fetal death is when the mother is infected during the first half (20 weeks) of pregnancy, after which the excess risk goes to zero. But 80% of fetal deaths occur within 4 weeks and almost all within 12 weeks of maternal infection (Enders et al. 2004). The lines 457-458 may thus need to be slightly modified, unless more proof is given of first-trimester fetal infection affecting the fetus more severely than middle-trimester fetal infection, i.e., differentiating maternal and fetal infections. Of course this may be true, because the early fetus is generally more vulnerable, but it would help the case if some proof is provided from the literature.

18: Line 423-432 and 471-478: It is stated that globoside is used as B19V receptor in acidic conditions (airways) and VP1uR in neutral conditions, such as bone marrow. However, no pH measurements have been done in this study for the placental cells? Normally the placenta is known to be slightly alkaline, which corroborates the use of VP1uR, yet only globoside was studied in the human placenta, but no pHs, so please clarify why and add placental pH aspect to the Discussion.

19: Line 449: “Our findings indicate that both CTB and STB populations are susceptible to B19V.” Please elaborate which exact cells are here referred to and which experiments indicate this. Also, clarify in the text if “susceptible to B19V” here means replication after B19V infection or only transcription after B19V-DNA transfection?

20. Line 459: The highly proliferative BeWo and JEG-3 “which originate from first-trimester trophoblasts” could also be fast growing due to malignancy, and not for being from the first trimester. AS mentioned, they are not physiologically relevant as model cells. This would be good to be discussed.

21: Line 480-481: “this study demonstrates that B19V can enter and infect placental trophoblasts”: This study actually shows on one hand, that the virus genome can be transfected into some immortal placental cells leading to transcription and translation, and on the other hand that that virions can enter some but not all cells, but to be strict, it does not really show that the B19 virus can infect, transcribe, replicate and produce progeny viruses in true placental trophoblasts. If really not able to infect those natural placental cells, it would be good to at least explain why 1+1 must be 2.

22: There is a lot of methodology included in the Results section, but it still does not include everything, so please refer in Results to the Materials and Methods section if more details are given there, and vice versa, include in short under the Materials and Methods subheadings, which experiment used which method.

Reviewer #3: To increase the readability of the paper the cells lines mentioned in the text should be in the same order as in the figures e.g. Fig 1, Fig 3,

The results of the supplementary Figure S1 A, B, and C are not adequately explained in the results.

In Fig 2B and Fig 3B the multinucleated syncytia formation of hPTCCTB cells due to STB differentiation with forskolin is not visible from the images. This should be shown, as syncytia formation is used as a marker for successful differentiation.

The beginning of Figure 4 legend contains text that should be in the Materials and Methods section. Please modify.

In Figure 7E. The amount of VP1/VP2 mRNA in BeWo cells at 24 and 48 hpi is 0 in the absence of PEI. This is confusing when compared to Fig. 4. Could you please explain in detail?

Fig.8C, E. The text on top of the diagrams is too small.

PLOS authors have the option to publish the peer review history of their article (what does this mean? ). If published, this will include your full peer review and any attached files.

**Do you want your identity to be public for this peer review?** For information about this choice, including consent withdrawal, please see our Privacy Policy .

Reviewer #1: No

Reviewer #2: No

Reviewer #3: No

**Figure resubmission:**

**Reproducibility:**



---

## [Editor Report · Decision Letter 1]

9 Feb 2026

Dear Dr. Ros,

We are pleased to inform you that your manuscript 'Cellular determinants of parvovirus B19 susceptibility in the human placenta' has been provisionally accepted for publication in PLOS Pathogens.

Best regards,

Kinjal Majumder, PhD

Guest Editor

PLOS Pathogens

Donna Neumann

Section Editor

PLOS Pathogens

Sumita Bhaduri-McIntosh

Editor-in-Chief

PLOS Pathogens

orcid.org/0000-0003-2946-9497

Michael Malim

Editor-in-Chief

PLOS Pathogens

orcid.org/0000-0002-7699-2064
---

## [Editor Report · Acceptance letter]

Dear Dr. Ros,

We are delighted to inform you that your manuscript, "

Cellular determinants of parvovirus B19 susceptibility in the human placenta," has been formally accepted for publication in PLOS Pathogens.

Best regards,

Sumita Bhaduri-McIntosh

Editor-in-Chief

PLOS Pathogens

orcid.org/0000-0003-2946-9497

Michael Malim

Editor-in-Chief

PLOS Pathogens

orcid.org/0000-0002-7699-2064